# Immunoproteasome functions explained by divergence in cleavage specificity and regulation

Michael B Winter[1][†][§], Florencia La Greca[1][†], Shirin Arastu-Kapur[1,2][†], Francesco Caiazza[1], Peter Cimermancic[3][#], Tonia J Buchholz[2], Janet L Anderl[2], Matthew Ravalin[1], Markus F Bohn[1], Andrej Sali[1,3], Anthony J O'Donoghue[4][‡]*, Charles S Craik[1][‡]*

[1]Department of Pharmaceutical Chemistry, University of California, San Francisco, San Francisco, United States; [2]Onyx Pharmaceuticals, Inc., an Amgen subsidiary, San Francisco, United States; [3]Department of Bioengineering and Therapeutic Sciences, California Institute for Quantitative Biosciences, University of California, San Francisco, San Francisco, United States; [4]Skaggs School of Pharmacy and Pharmaceutical Sciences, University of California, San Diego, San Diego, United States

*For correspondence:
ajodonoghue@ucsd.edu (AJO'D);
charles.craik@ucsf.edu (CSC)

[†]These authors contributed equally to this work
[‡]These authors also contributed equally to this work

Present address: [§]CytomX Therapeutics, South San Francisco, United States; [#]Verily Life Sciences, South San Francisco, United States

**Abstract** The immunoproteasome (iP) has been proposed to perform specialized roles in MHC class I antigen presentation, cytokine modulation, and T cell differentiation and has emerged as a promising therapeutic target for autoimmune disorders and cancer. However, divergence in function between the iP and the constitutive proteasome (cP) has been unclear. A global peptide library-based screening strategy revealed that the proteasomes have overlapping but distinct substrate specificities. Differing iP specificity alters the quantity of production of certain MHC I epitopes but does not appear to be preferentially suited for antigen presentation. Furthermore, iP specificity was found to have likely arisen through genetic drift from the ancestral cP. Specificity differences were exploited to develop isoform-selective substrates. Cellular profiling using these substrates revealed that divergence in regulation of the iP balances its relative contribution to proteasome capacity in immune cells, resulting in selective recovery from inhibition. These findings have implications for iP-targeted therapeutic development.
DOI: https://doi.org/10.7554/eLife.27364.001

## Introduction

Mammalian proteasomes are complexes with multiple protease active sites that carry out the regulated degradation of proteins, mediating processes such as the clearance of mutated or misfolded proteins, cell signaling, and antigen presentation (*Kisselev et al., 2012*; *Stadtmueller and Hill, 2011*). The 20S proteasome core particle is a barrel-shaped structure composed of four stacked heptameric rings (*Tomko and Hochstrasser, 2013*). The two outer rings, containing the α-subunits, act as binding sites for regulatory complexes (e.g., 19S, 11S, and PA200 families) that stimulate its peptidase activity and modulate access of cytosolic proteins into the proteasome inner chamber (*Stadtmueller and Hill, 2011*; *Tomko and Hochstrasser, 2013*). The two inner rings contain the β-subunits, three of which (β1, β2, and β5) have an active site containing a catalytic threonine residue (*Tomko and Hochstrasser, 2013*). These catalytic subunits have been traditionally referred to as "caspase-," "trypsin-," and "chymotrypsin-like" (CT-L), respectively, based on their general substrate specificity preferences (*Kisselev et al., 2012*).

In addition to the constitutive proteasome (cP), which is expressed in all cell types, an immuno-proteasome (iP) isoform is predominantly expressed in cells of hematopoietic origin (**McCarthy and Weinberg, 2015**). The overall architecture of the iP and cP core particles is identical; however, the iP contains different catalytic subunits – referred to as LMP2 (iβ1), MECL-1 (iβ2), and LMP7 (iβ5) – that share respectively 62%, 59%, and 71% sequence identity with their cP counterparts (**Ferrington and Gregerson, 2012**). Expression of all immuno catalytic subunits, along with the 11S-type activator PA28, is upregulated by pro-inflammatory signals, such as IFN-γ, during the immune response. Mapping of the LMP2 (*PSMB9*) and LMP7 (*PSMB8*) genes to the major histocompatibility complex (MHC) locus combined with interferon-induced activation first led to the proposal that the iP carries out a specialized role in generating peptides for MHC class I antigen presentation (**Ferrington and Gregerson, 2012**). Furthermore, early biochemical evidence suggested that the iP catalytic subunits show an enhanced ability to generate peptides with specific C-terminal residues that may better match the terminal (anchor) amino acid residue preferences of MHC I molecules (**Gaczynska et al., 1993**; **Driscoll et al., 1993**; **Gaczynska et al., 1994**; **Cardozo and Kohanski, 1998**; **Toes et al., 2001**). More recently, it was found that mice completely lacking in iP catalytic subunits display an antigen repertoire that differed from wild-type mice and resulted in transplant rejection (**Kincaid et al., 2011**). These findings have suggested a possible role for iP cleavage specificity in shaping the immune response.

Inhibition of proteasome CT-L activity was initially pursued as a therapeutic strategy for the treatment of cancers with highly proliferative and proteasome-dependent cells; however, current FDA-approved proteasome inhibitors lack selectivity between the cP and iP (**Kisselev et al., 2012**; **Adams, 2004**). Bortezomib (Velcade), a reversible boronic acid-based inhibitor, provided clinical validation of proteasome inhibition in multiple myeloma (**Richardson et al., 2003**). The second-generation proteasome inhibitor, carfilzomib (CFZ; Kyprolis or PR-171), which also targets the CT-L subunits, uses an irreversible epoxyketone warhead and shows reduced off-target inhibition, while circumventing resistance against bortezomib (**Arastu-Kapur et al., 2011**; **Siegel et al., 2012**). Recently, considerable effort has been dedicated toward the development of iP-selective inhibitors (**Johnson et al., 2017**; **Dubiella et al., 2015**; **Sosič et al., 2016**; **Muchamuel et al., 2009**) in the interest of selectively targeting iP-dominant cells for the treatment of autoimmune disorders and certain cancers. In particular, the epoxyketone inhibitor, ONX 0914 (PR-957), which preferentially inhibits the LMP7 subunit, was able to block cytokine production and attenuate disease progression in a rheumatoid arthritis mouse model at a significantly lower dose than bortezomib or CFZ (**Muchamuel et al., 2009**). In addition, ONX 0914 has showed promise for the treatment of lupus and multiple sclerosis (**Ichikawa et al., 2012**; **Basler et al., 2014**) and for inflammation-associated colorectal cancer (**Koerner et al., 2017**; **Vachharajani et al., 2017**).

Although the iP has emerged as a promising therapeutic target, it is not well understood how its functions diverge from those of the cP, which is often found within the same cell. Here, we applied a recently-developed peptide library-based profiling strategy (**O'Donoghue et al., 2012**; **O'Donoghue et al., 2015**) to define the substrate specificities of each proteasome in a global and an unbiased manner. This analysis revealed that the iP and cP have overlapping substrate specificities, with some significant differences. These differences were exploited to rationally design LMP7- and β5-selective fluorogenic peptide substrates for the cellular profiling of each proteasome. The global substrate specificities of the iP and cP were then compared to MHC I peptides and further queried experimentally using a synthetic peptide library designed from amino acid sequences flanking MHC I peptide-processing sites. This analysis suggested that the specificity of the iP may alter the extent of display of certain MHC I peptides but does not appear to be preferentially suited for MHC I epitope generation. Furthermore, an evolutionary analysis suggested that divergent iP substrate specificity is due to genetic drift from the cP. Finally, the iP- and cP-selective fluorogenic substrates were used to probe the contribution of the iP to ubiquitin-proteasome system (UPS) capacity in immune cells and assess proteasome recovery (or 'bounce-back') (**Radhakrishnan et al., 2010**; **Sha and Goldberg, 2014**; **Radhakrishnan et al., 2014**; **Weyburne et al., 2017**) from iP-selective inhibitor treatment. Our results suggest that divergence in iP substrate specificity primarily impacts the content of the immune repertoire by altering the quantity of production of certain epitopes and that divergence in iP regulation balances the contribution of each proteasome to the total UPS capacity.

## Results

### Global substrate specificity screen of the iP and cP

To provide biochemical insight into the substrate specificities of the iP and cP, we performed a global substrate screening approach referred to as Multiplex Substrate Profiling by Mass Spectrometry (MSP-MS) (*Figure 1*) (*O'Donoghue et al., 2012*; *O'Donoghue et al., 2015*). The MSP-MS assay uses an equimolar mixture of 228 rationally-designed, 14-mer peptide substrates and provides an unbiased assessment of protease substrate specificity. The iP or cP was added to the peptide library and peptide cleavage products were identified after 60, 120, 240, and 480 min of incubation through liquid chromatography-tandem mass spectrometry (LC-MS/MS). Only cleavage sites found in two biological replicates of each proteasome are reported. The proteasomes were found to have broad substrate specificity, being capable of hydrolyzing 391 (iP) and 330 (cP) different peptide bonds after 480 min of incubation; this corresponds to 13.2% and 11.1% of the total cleavage sites in the library for the iP and cP, respectively. Saturation of new cleavage-site usage was evident by the 480 min assay time point (*Figure 1—figure supplement 1*), reflecting depletion of the most favored cleavage sites in the library, and the ability of the MSP-MS assay to readily detect the most dominant cleavage events, as described in *Figure 1—figure supplement 2*.

Statistical analysis of both cleaved and uncleaved positions in the peptide library was performed to represent the fold enrichment and de-enrichment of amino acid residue types at the four sub-sites on both sides of the scissile bond (*Colaert et al., 2009*). This analysis revealed that the iP and cP have similar overall substrate specificity motifs (*Figure 1A*). Specificity is dominated by non-prime-side residues, in particular the P1 position, with a shared enrichment of extended hydrophobic residues at P1, P3 and P4 as well as lysine at P2. The preferred specificity for hydrophobic residues and arginine in the P1 position suggest that the CT-L subunits (β5 and LMP7) and trypsin-like subunits (β2 and MECL-1) cleave many more peptide bonds than the β1 and LMP2 subunits of the cP and iP, respectively. This is particularly clear from the lack of aspartic acid enrichment in the P1 position.

Comparison of individual peptide cleavage events within the MSP-MS library demonstrated that the iP and cP also have differential substrate specificity with 32.1% and 19.5% non-overlapping cleavage sites at 480 min for the iP and cP, respectively (*Figure 1B*, *Figure 1—figure supplement 1*, and *Supplementary file 1* for additional time points and reproducibility). To quantify global differences in substrate specificity, a difference map was generated using Z-scores derived from residue preferences at each sub-site (*Figure 1C* and *Figure 1—figure supplement 3* for additional time points) (*Colaert et al., 2009*). This analysis revealed significant physicochemical differences in P1 specificity. The iP displayed an increased preference for cleavage following bulky, hydrophobic amino acid residues such as tryptophan. In contrast, the cP displayed an increased preference for cleavage after basic residues such as arginine in addition to polar and smaller amino acid residues such as serine, threonine, glutamine, glycine, and alanine (*Figure 1—figure supplement 3* and *Figure 1—figure supplement 4*).

### Rational design of fluorogenic substrates with LMP7 and β5 selectivity

Using the specificity differences identified in the peptide degradation assay, we sought to develop chemical tools (*de Bruin et al., 2016*; *Dubiella et al., 2016*) that are substrate-based for readily monitoring LMP7 and β5 activities in cellular lysate conditions because these subunits represent a major activity of both proteasomes. To do this, we identified individual peptides in the MSP-MS library that displayed selectivity for the iP or cP and reflected hydrophobic specificity at the P1 position (*Figure 2A*). For LMP7, an internally-quenched (IQ) fluorogenic substrate was synthesized that corresponded to the P5 to P2' residues surrounding a Trp-Pro bond in an individual peptide. This substrate was designed to evaluate an LMP7-selective probe containing prime-side residues but proved to be non-selective for the iP when in IQ substrate format (*Figure 2B*). However, a shorter tetrapeptide sequence (P4-P1) bearing a 7-amino-4-carbamoylmethylcoumarin (*ACC*) fluorophore at the P1' position, EWHW-*ACC*, afforded 4.5-fold iP selectivity at 10 μM substrate concentration (*Figure 2B*). An additional tetrapeptide, PDFY-*ACC*, which shares no common amino acid residues with EWHW-*ACC*, was also synthesized. The PDFY-*ACC* substrate was preferentially cleaved by the iP, albeit with reduced activity and selectivity (2.2-fold) (*Figure 2B*). Taken together, these initial

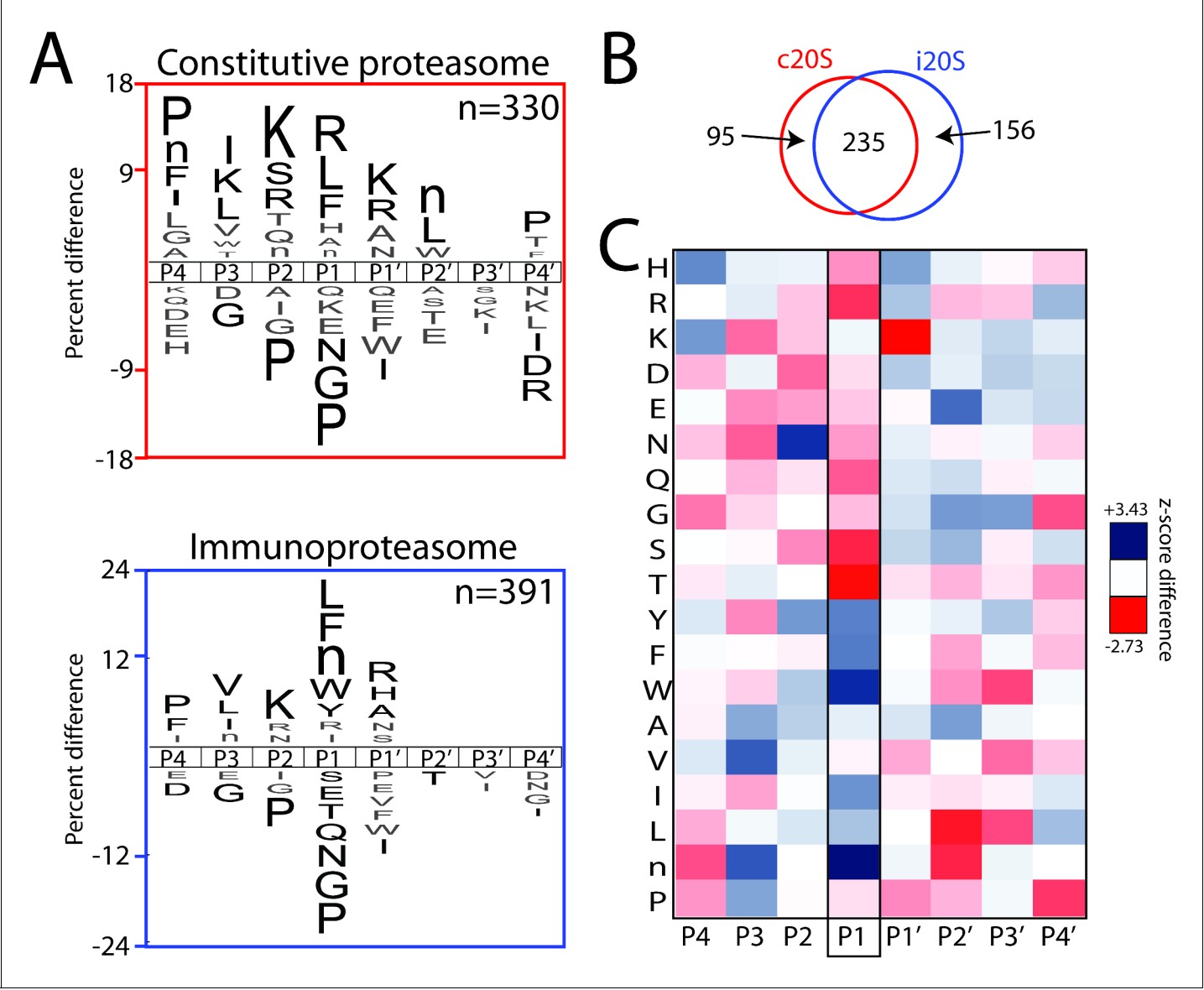

**Figure 1.** Global substrate specificity profiling of the iP and cP with Multiplex Substrate Profiling by Mass Spectrometry (MSP-MS) reveals shared and differential substrate specificity features. (**A**) iceLogo representations of iP and cP substrate specificity (P4–P4') at the 480 min assay time point (p≤0.05 for non-grayed residues (*Colaert et al., 2009*); 'n' is norleucine). (**B**) Quantification of the total shared and non-overlapping iP- and cP-derived cleavages in the peptide library at the 480 min assay time point. Venn diagrams for additional assay time points are provided in *Figure 1—figure supplement 1*, demonstrating a time-dependent increase in cleavage overlap. (**C**) Heat map representation of iP and cP specificity differences using Z-scores (*Colaert et al., 2009*) calculated for the P4-P4' positions. Differences in P1 specificity are highlighted. Heat maps for additional time points are provided in *Figure 1—figure supplement 3*. MSP-MS revealed that the iP has an increased preference for certain bulky, hydrophobic amino acid residues at the P1 position, whereas the cP has an increased preference for smaller and polar amino acid residues at the P1 position. A qualitative comparison of the P1 specificity differences identified using the MSP-MS library with those reported in *Toes et al., 2001* and *Mishto et al., 2014* is provided in *Figure 1—figure supplement 4*. Two biological replicates were assayed for each proteasome and only overlapping cleavages between replicates are reported. An analysis of biological (*Figure 1—figure supplement 1*) and technical (*Supplementary file 1*) reproducibility is provided.
DOI: https://doi.org/10.7554/eLife.27364.002

The following source data and figure supplements are available for figure 1:

**Source data 1.** The following are contained in a supplementary file for the MSP-MS assay: a sample key; the sequences of the peptide library in FASTA format; a full mass spectrometry peptide report; processed cleavage data showing spectral counts for each octapeptide (**P4–P4'**) and the associated cleavage within the parental peptide; processed cleavage data (indicated as 'negative') with cleavages also appearing in the no-enzyme control (NEC) highlighted; all octapeptides (**P4–P4'**) to be used as the background dataset in iceLogo; 'positive' octapeptide (**P4–P4'**) cleavages with the 'negative'

*Figure 1 continued on next page*

*Figure 1 continued*

cleavage data removed; 'accumulative' octapeptide (**P4**–**P4'**) cleavages at each assay time point for the proteasome biological replicates; a comparison of cleavages common between biological replicates; and heat maps based on Z-scores showing the specificity at each assay time point.

DOI: https://doi.org/10.7554/eLife.27364.007

**Figure supplement 1.** Comparison of iP- and cP-derived peptide cleavage products across all time points in the MSP-MS assay.

DOI: https://doi.org/10.7554/eLife.27364.003

**Figure supplement 2.** The MSP-MS assay detects the most dominant peptide cleavage events.

DOI: https://doi.org/10.7554/eLife.27364.004

**Figure supplement 3.** Heat maps for the iP and cP across multiple MSP-MS assay time points demonstrating shared and differential specificity features at the P4 to P4' positions.

DOI: https://doi.org/10.7554/eLife.27364.005

**Figure supplement 4.** Qualitative comparison of the major differences in human iP and cP P1 specificity preferences identified using the MSP-MS library compared to those reported in previous profiling studies.

DOI: https://doi.org/10.7554/eLife.27364.006

studies indicated that short peptide sequences identified in the MSP-MS assay could be optimized to develop substrates with improved iP selectivity.

To identify candidate residue positions for rational sequence optimization, both proteasomes were assayed using Positional Scanning-Synthetic Combinatorial Libraries (PS-SCLs) to further assess non-prime-side specificity preferences at the P4, P3, and P2 positions (*Figure 2—figure supplement 1*) (*Harris et al., 2000*). The PS-SCLs are divided into pooled sub-libraries of tetrapeptides (160,000 total) containing a fixed amino acid residue at a given position, an isokinetic mixture of 20 amino acid residues at each remaining site, and an ACC fluorophore at the P1' position. Cleavage sensitivity to CFZ pretreatment was used to probe LMP7 or β5 specificity at each site (*Figure 2—figure supplement 2*). To improve the iP selectivity of the EWHW-*ACC* parent substrate, select amino acid residues were integrated within the sequence (*Figure 2B* and *Figure 2—figure supplement 3*) based on amino acid preferences identified using both profiling approaches. Among the residues queried, a P2 phenylalanine substitution (EWFW-*ACC*) resulted in the most dramatic improvement in both specific activity and iP selectivity. Supporting Michaelis-Menten analysis further demonstrated that EWFW-*ACC* had 7-fold improvement in $K_m$ for the iP with 16-fold selectivity at $V_{max}$ compared to the 6-fold selectivity of the parent substrate (*Figure 2C* and *Supplementary file 2*).

To develop a fluorogenic substrate with β5 selectivity, an MSP-MS peptide cleavage with cP selectivity (*Figure 2A*) was used to synthesize an initial tetrapeptide ACC substrate, SHRn-*ACC* (*Figure 2D*). However, this substrate had only modest cP selectivity (*Figure 2D*). An existing in-house substrate (YVQA-*ACC*), bearing P1 alanine as well as the cP-favored P2 glutamine was evaluated. This substrate was found to afford similar selectivity (1.8-fold) but higher cP activity (*Figure 2D*). Modifications to the YVQA-*ACC* sequence with cP-favored residues, such as P1 serine and glycine, did not offer improvements in selectivity. Therefore we evaluated the effect of modifying the peptide length. A tripeptide bearing the sequence VQA-*ACC* was found to have dramatically reduced activity but increased cP selectivity (3.0- versus 1.8-fold selectivity at 10 µM substrate concentration) (*Figure 2D*). Next, *N*-terminal capping groups were screened and found to greatly improve activity of the tripeptide substrate. The 5-methylisoxazolyl (iso) group utilized in the β5-favored inhibitor, PR-825, yielded both the highest cP specific activity and selectivity (4.4-fold), suggesting that the iso-VQA-*ACC* probe was a promising lead candidate (*Figure 2D* and *Figure 2E*). Selectivity of the EWFW-*ACC* and iso-VQA-*ACC* substrates was retained under varied proteasome activation conditions (*Figure 2—figure supplement 4*).

## Fluorogenic substrates enable cellular activity profiling of LMP7 and β5

To confirm the respective selectivity of the EWFW-*ACC* and iso-VQA-*ACC* probes for the iP and cP under more complex assay conditions, activity assays were performed using defined mixtures of each proteasome (*Figure 3A*). Both probes retained selectivity for their target proteasomes under conditions of iP and cP competition, as evidenced by direct correlation of probe activity with the level of the respective target proteasome. In contrast, the commercial β5/LMP7 substrate, LLVY-*AMC*, was unable to distinguish between the two proteasomes when assayed at a comparable substrate concentration (*Figure 3A* and *Figure 3—figure supplement 1*). Probe efficacy was further

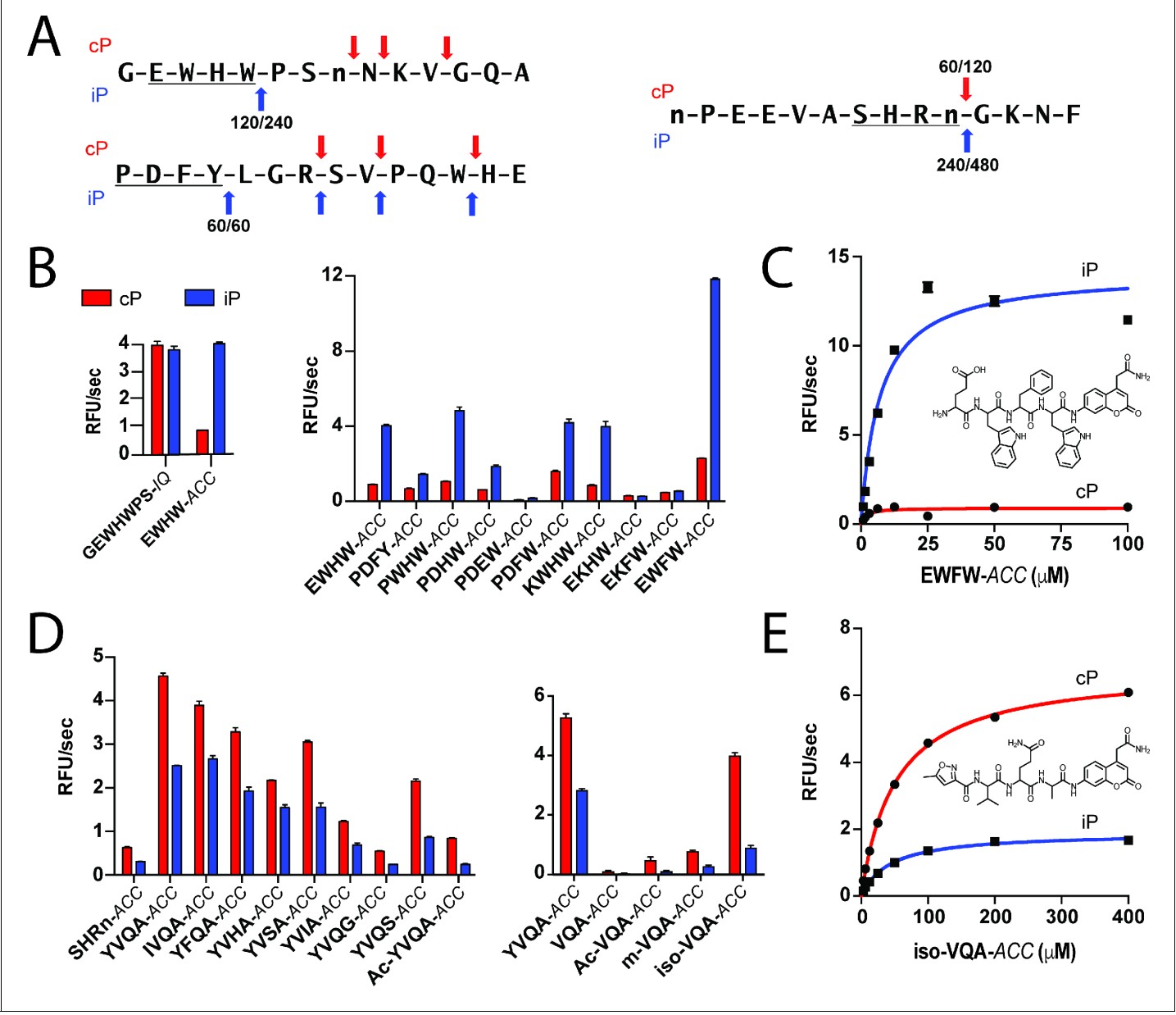

**Figure 2.** Rational optimization of tetrapeptide ACC substrates with iP and cP selectivity. (A) Example 14-mer peptides from the iP and cP MSP-MS assays associated with **Figure 1** illustrating cleavages that were used as a template for rational fluorogenic substrate design. All cleavages common between two biological replicates of each proteasome are reported with the time point of first appearance noted for each replicate. (B) Selectivity of parental IQ and tetrapeptide ACC fluorogenic substrates derived from an iP-favored cleavage (**Figure 2A**, top) in the MSP-MS library. Subsequent rational substrate optimization is shown. (C) Michaelis-Menten characterization and chemical structure of the lead iP substrate, EWFW-*ACC*. (D) Optimization of an ACC substrate with cP selectivity using a similar approach. Shortening the peptide sequence to a tripeptide and addition of an N-terminal capping group were found to be important for improving substrate selectivity and maintaining specific activity. (E) Michaelis-Menten characterization and chemical structure of the lead cP substrate iso-VQA-*ACC*. For all fluorogenic substrate assays, mean activity is reported with error bars representing the standard deviation from $n$ = 3 replicates.

DOI: https://doi.org/10.7554/eLife.27364.008

The following figure supplements are available for figure 2:

**Figure supplement 1.** P4-P2 substrate specificity of SDS-activated iP and cP using the PS-SCL profiling approach.

DOI: https://doi.org/10.7554/eLife.27364.009

**Figure supplement 2.** PS-SCL profiling of SDS-activated iP and cP in the absence and presence of 40 nM CFZ pretreatment.

DOI: https://doi.org/10.7554/eLife.27364.010

**Figure supplement 3.** Synthetic approach for peptide substrates bearing a C-terminal ACC fluorophore.

*Figure 2 continued on next page*

*Figure 2 continued*

DOI: https://doi.org/10.7554/eLife.27364.011

**Figure supplement 4.** Comparison of fluorogenic substrate selectivity under SDS (0.03%) and PA28 (12 eq.) activation conditions.

DOI: https://doi.org/10.7554/eLife.27364.012

evaluated in clinically relevant cell lysates from whole blood, a T-lymphoblast cell line (MOLT-4), and peripheral blood mononuclear cells (PBMCs), which contain varied proteasome ratios (*Figure 3A*). Relative substrate cleavage rates correlated with the ratios of active LMP7 to β5 in each cell lysate (1:99, 43:57, and 100:0, respectively), as determined using the ProCISE assay, which relies on direct active site labeling and ELISA measurement (*Figure 3—figure supplement 2*) (*Parlati et al., 2009*). To confirm iso-VQA-*ACC* and EWFW-*ACC* selectivity for the β5 and LMP7 subunits, the lysates underwent pretreatment with 100 nM CFZ or the LMP7- and β5-selective inhibitors, ONX 0914 and PR-825, respectively (*Muchamuel et al., 2009*). Substrate cleavage in the cell lysates correlated with the expected pattern of inhibition (*Figure 3B*); further dose-response curves using the MOLT-4 lysate (*Figure 3C*) yielded IC$_{50}$ values that matched those obtained from the ProCISE assay, demonstrating the isoform selectivity of the substrates even at the approximately 50:50 proteasome level (*Supplementary file 3*). In contrast, a commercial substrate (Ac-WLA-*AMC*) commonly used to measure β5 activity was unable to specifically detect proteasome activity using this lysate condition (*Figure 3—figure supplement 3*). Together, these activity-profiling results suggest that the fluorogenic substrates can selectively monitor LMP7 and β5 activity in lysates and that similar CT-L substrate specificity determinants operate under cellular conditions.

## Substrate specificity of the iP primarily alters MHC I peptide cleavage efficiency

Our global assessment of iP and cP peptide degradation patterns revealed that these enzymes have similar substrate specificities, with some clear differences. To expand the scope of our analysis, we sought to address the extent to which the substrate specificities of the iP and cP shape the repertoire of peptides displayed during MHC class I antigen presentation. Using the substrate specificity profiles from the MSP-MS assay, we initially applied a bioinformatic approach to score MHC I peptides based on their relative propensity for cleavage by either proteasome (*Figure 4—figure supplement 1*). Peptide scoring was performed using a recently reported large proteomic dataset of MHC I peptides (*n* = 22,598) that were displayed by allelic-diverse MHC I complexes in multiple human cell lines (*Bassani-Sternberg et al., 2015*). In this dataset, the most frequently occurring C-terminal amino acid residues were the long-chain residues leucine (27.6%), lysine (19.1%), and tyrosine (10.8%). In our specificity differentiation profiles (*Figure 1C* and *Figure 1—figure supplement 3*), the iP and cP had an overall similar preference for cleavage on the C-terminal side (P1 position) of these residues, particularly lysine, which indicates that many MHC I peptides have a similar potential of being generated by either proteasome (*Figure 4—figure supplement 1*). Peptides with C-terminal tryptophan are much more likely to be generated by the iP relative to the cP but were observed in only 0.30% of MHC I peptides. Similarly, peptides with smaller and polar C-terminal residues are more likely to be generated by the cP and were also less frequently observed (e.g., 6.0% for alanine and less than 0.4% for serine, threonine, and other cP-favored amino acids). Notably, comparison of MHC I C-terminal amino acid residue distribution to the human proteome revealed de-enrichment of these preferential P1 residues (*Figure 4—figure supplement 1*), suggesting that their propensity for display by MHC I molecules might be reduced compared to residues that are similarly preferred by both proteasomes.

To provide an experimental measure of iP and cP cleavage selectivity for MHC I peptides, we constructed a synthetic peptide library consisting of the 7 amino acid residues flanking sites of MHC I processing, using sequences selected at random from the proteomic data set. The peptides in the library were pooled at equimolar concentration and incubated with the iP or cP. Peptide cleavage products were identified over time using LC-MS/MS, and label-free quantitation was used to calculate relative rates of peptide hydrolysis (between positions 7 and 8 at the MHC I site). In agreement with our bioinformatic approach, the proteasomes hydrolyzed most peptides at a similar rate ($\leq 2$ fold difference) (*Figure 4A*). However, several peptides were clearly favored by either the cP or iP

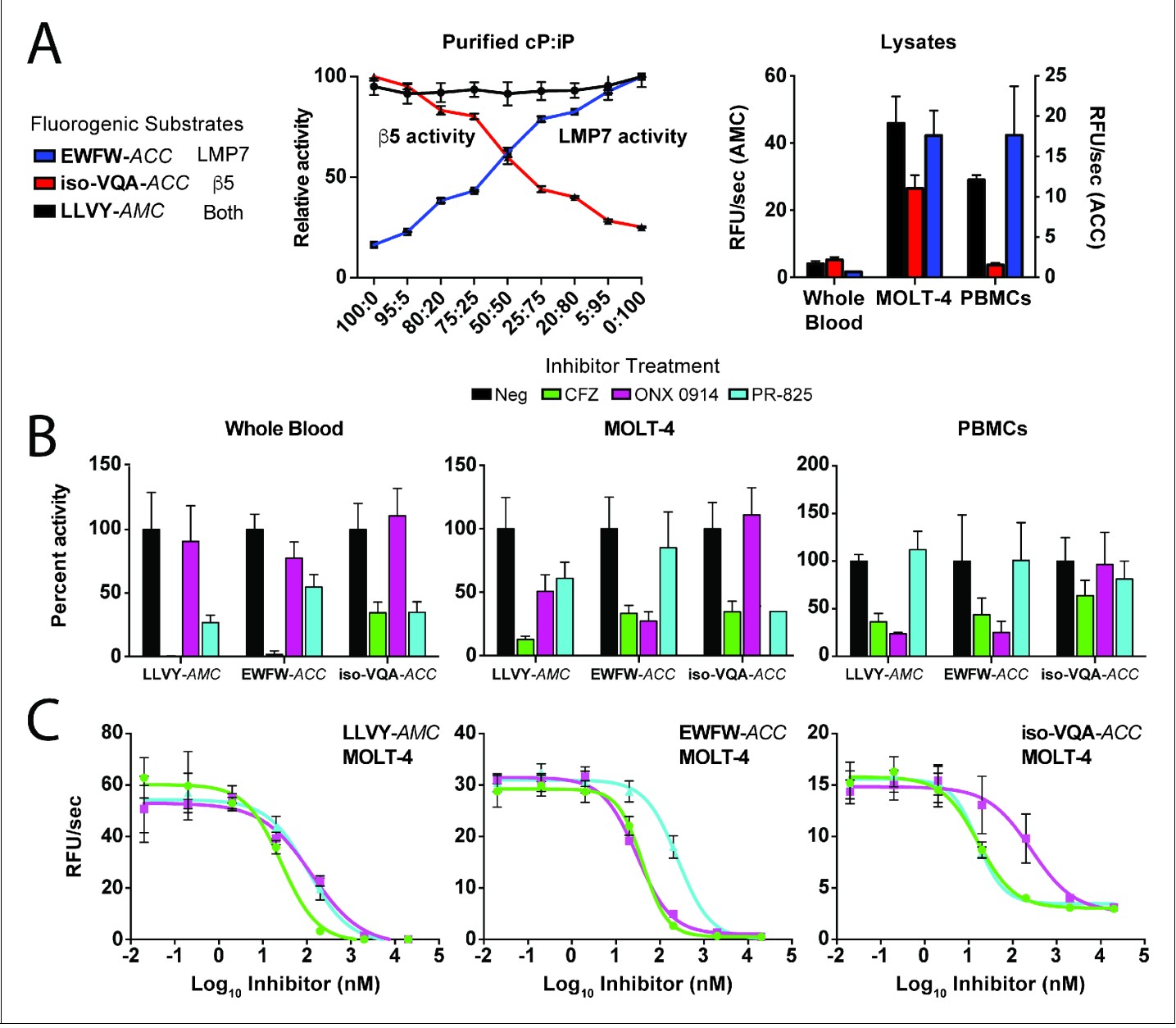

**Figure 3.** Optimal fluorogenic substrates enable the selective monitoring of LMP7 and β5 activity in cell lysates containing varied proteasome ratios. (**A**) Demonstration of EWFW-*ACC* (10 μM) and iso-VQA-*ACC* (30 μM) selectivity for the iP and cP, respectively, compared to the universal substrate LLVY-*AMC* (30 μM) using mixed ratios of purified proteasomes. A comparison is provided of EWFW-*ACC* and iso-VQA-*ACC* specificity in whole blood, MOLT-4, and PBMC lysates. (**B**) Pretreatment of the lysates with 100 nM CFZ, an LMP7-favored inhibitor (ONX 0914), or a β5-favored inhibitor (PR-825) demonstrates the ability of the substrates to distinguish their target proteasomes under lysate conditions. (**C**) IC$_{50}$ determination for CFZ, ONX 0914, and PR-825 in the mixed LMP7/β5 MOLT-4 lysate using LLVY-*AMC* and the selective substrates. For all assays, mean activity is reported with error bars representing the standard deviation from $n$ = 3 replicates.

DOI: https://doi.org/10.7554/eLife.27364.013

The following figure supplements are available for figure 3:

**Figure supplement 1.** Michaelis-Menten plot of the iP and cP with the commercial LLVY-*AMC* substrate.

DOI: https://doi.org/10.7554/eLife.27364.014

**Figure supplement 2.** Quantification of active LMP7 and β5 levels in lysates from whole blood, the MOLT-4 cell line, and PBMCs through direct active site labeling using the ProCISE assay.

DOI: https://doi.org/10.7554/eLife.27364.015

*Figure 3 continued on next page*

*Figure 3 continued*

**Figure supplement 3.** Improved ability of the iso-VQA-*ACC* substrate to detect β5 activity in the MOLT-4 lysate compared to a commercially available β5 fluorogenic substrate.

DOI: https://doi.org/10.7554/eLife.27364.016

with up to 12-fold and 15-fold selectivity, respectively. Among the majority of cleavages with P1 specificity for hydrophobic residues, the sequences HTQVIELERKFSHQ and VSEGTHFLETIETP displayed the greatest cP and iP selectivity, respectively. As expected, the peptide substrate containing a P1 tryptophan, KNTFPKWKPGSLAS, was preferentially cleaved by the iP (*Figure 4B*). Taken together, our bioinformatic and substrate profiling approaches revealed that the majority of MHC I peptides have a similar potential for being generated by either proteasome. Differing iP specificity appears to impact the content of the MHC I repertoire primarily by altering the quantity of production of certain epitopes. We note that this outcome appears to be only partly explained by the differential preference of the iP for certain P1 residues.

## Divergence of LMP7 specificity likely occurred by neutral evolution

Because of the lack of preferential correlation of iP specificity with MHC I C-terminal residue preferences, we sought to provide an evolutionary basis for the divergence in iP and cP cleavage specificities. Notably, the 14 subunits of the cP are more ancient proteins than those of the iP; the catalytic iP subunits first emerged through gene duplication in jawed vertebrates along with other components of the immune system (*Kesmir et al., 2003*). To characterize potential differences in evolutionary selection of the iP and cP, we compared LMP7 and β5 sequence conservation across organisms that contain both proteasomes (*Sutoh et al., 2012*). We then estimated the balance between neutral mutations, purifying selection, and positive selection through comparison of rates of nonsynonymous (amino acid residue-changing) and synonymous (amino acid residue-preserving) nucleotide substitution (*Massingham and Goldman, 2005*). This analysis revealed that residues at LMP7 subunit interfaces and within the active site are highly varied among species and have undergone genetic drift. In contrast, the corresponding residues in the β5 subunit are highly conserved and have evolved through purifying selection (*Figure 4C*). These neutrally evolved residues in human LMP7 include Val31 in the S1 pocket, Cys48 (Gly in β5) in the S2 pocket, and Ser27 and Ala28 (Ala and Ser, respectively, in β5) in the S3 pocket and suggest lack of evolutionary pressure on iP specificity (*Figure 4—figure supplement 2*). Thus, neutral mutations may have helped diversify iP specificity in the presence of the rapidly evolving proteomes of diverse pathogens while preserving essential core recognition of host cell proteins.

## iP activity contributes to ups capacity and undergoes selective recovery from inhibition.

The importance of the iP in antigen presentation has been well appreciated; however, the contribution of the iP to cellular proteostasis and its regulation under basal (or non-immunostimulatory) conditions is not well understood. To confirm that iP activity has an impact on UPS capacity in immune cells under basal conditions, we used the selective substrates to identify a B-lymphoblast cell line (SUP-B15) (*Bassani-Sternberg et al., 2015*) with predominantly LMP7 activity (80:20). A pulse-treatment strategy was employed with ONX 0914 and PR-825 to evaluate the relative contribution of each active site to the turnover of ubiquitinated substrates under conditions with optimal inhibitor selectivity (*Muchamuel et al., 2009*). SUP-B15 cells were pulse-treated for one hour, and inhibitor selectivity was confirmed in cytosolic lysates up to three hours post-treatment through monitoring selective reduction of EWFW-*ACC* and iso-VQA-*ACC* cleavage (*Figure 5A*). Immunoblotting demonstrated global accumulation of ubiquitinated protein upon inhibition of LMP7 activity with ONX 0914, whereas no significant change in ubiquitination was observed upon inhibition of β5 activity with PR-825 (*Figure 5A* and *Figure 5—figure supplement 1*). This finding suggests that the iP makes a significant contribution to proteasome capacity in agreement with *Seifert et al., 2010* and that in SUP-B15 cells this contribution is correlated with the higher relative LMP7 activity level.

Because iP activity significantly contributes to basal UPS capacity, we next assessed whether immune cells upregulate proteasome activity as a recovery response to selective and irreversible

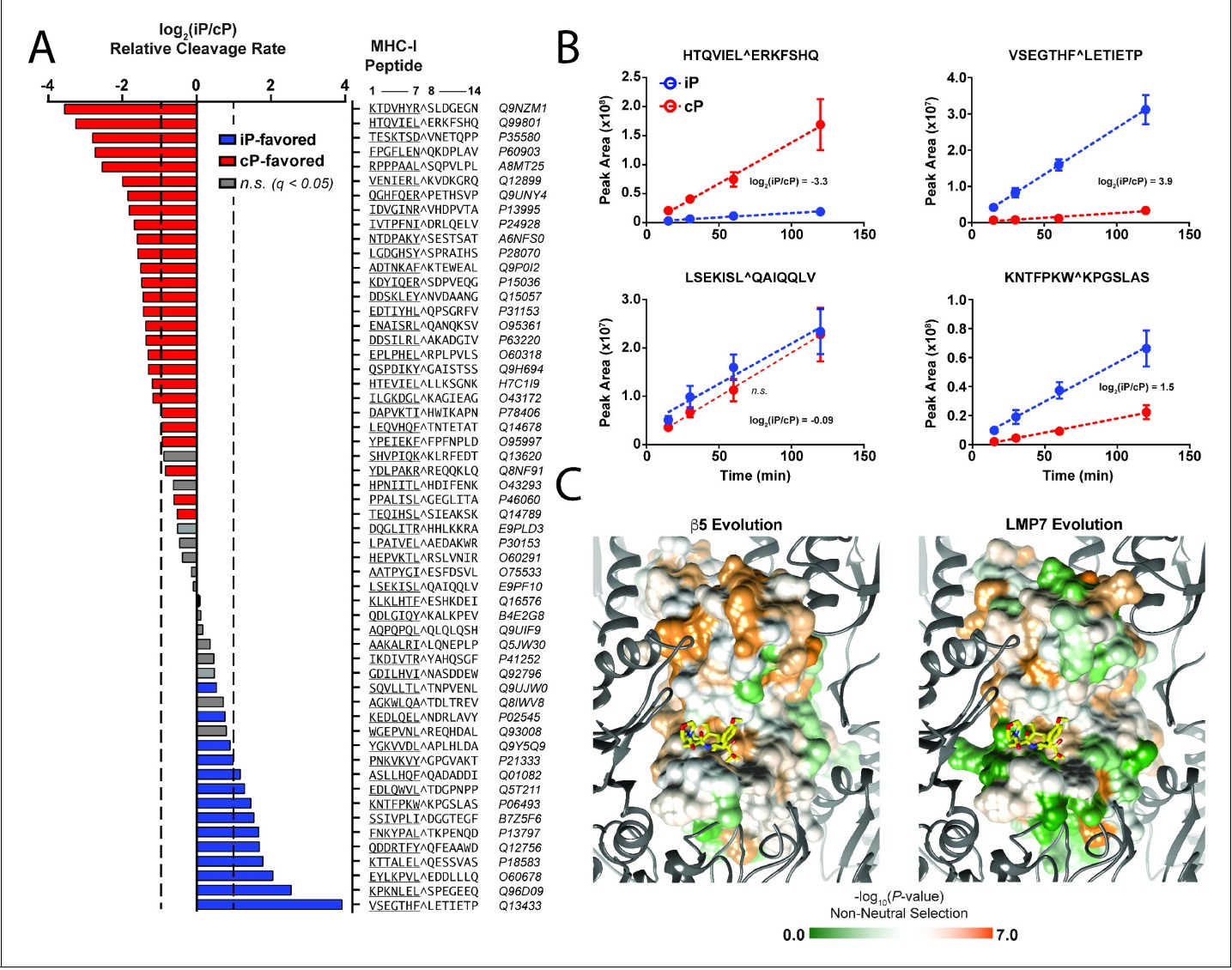

**Figure 4.** Differential substrate specificity of the iP alters MHC I peptide cleavage and is likely due to neutral evolution from the ancestral cP. (**A**) Label-free quantitation of relative iP and cP cleavage rates against a library of synthetic peptides derived from sequences flanking sites of MHC I peptide processing (**Bassani-Sternberg et al., 2015**). Residues corresponding to the C-terminal portion of a given MHC I peptide are at positions 1–7 in the peptide library, whereas residues corresponding to the subsequent parent protein sequence are at positions 8–14. Average relative cleavage rates ($\log_2$) are provided for iP and cP substrates between positions 7 and 8 (P1/P1') for all peptides in the library that underwent cleavage at this site and for which quantification in all replicates was possible ($n = 4$). Two-fold differences in relative cleavage rate ($\log_2 = 1$) are indicated with a dashed line. Non-grayed bars represent statistically significant ($q < 0.05$) differences in selectivity as determined using a Student's $t$-test (see Statistical methods). (**B**) Example kinetic traces from the MHC I peptide library time course showing cleavages following hydrophobic residues that have either high selectivity or no significant selectivity for the iP. Mean peak areas are reported with error bars representing the standard deviation ($n = 4$). (**C**) Evolutionary selection of residues in the β5 and LMP7 subunits across species that contain both proteasome isoforms. The β5 subunit has undergone more significant non-neutral evolution (residues are colored in orange) compared to the LMP7 subunit. Differences in evolutionary selection may account for the divergence in LMP7 and β5 cleavage specificity. Sequence alignments are provided in **Figure 4—figure supplement 2**.

DOI: https://doi.org/10.7554/eLife.27364.017

The following source data and figure supplements are available for figure 4:

**Source data 1.** The following are contained in a supplementary file for the MHC cleavage assay: a sample key; the sequences of the peptide library in FASTA format; and a full mass spectrometry peptide report.

DOI: https://doi.org/10.7554/eLife.27364.018

**Figure supplement 1.** Prediction of MHC I peptide cleavage by the iP and cP.

DOI: https://doi.org/10.7554/eLife.27364.019

**Figure supplement 2.** Protein sequence alignment for (**A**) β5 and (**B**) LMP7 in diverse species containing both proteasomes.

*Figure 4 continued on next page*

*Figure 4 continued*

DOI: https://doi.org/10.7554/eLife.27364.020

LMP7 inhibition. The SUP-B15 cell line was pulse-treated with ONX 0914 for one hour, and LMP7 and β5 activities were followed over an extended 24 hr time course. We observed that irreversible LMP7 inhibition in the SUP-B15 cell line was associated with a rapid recovery (or 'bounce-back') of LMP7 activity without an appreciable compensatory increase in the basal β5 activity level (*Figure 5B*), suggesting that iP activity is critical to proteasome capacity in this cell line and that cP activity may be differentially regulated. For comparison, we also subjected the MOLT-4 cell line, which has an approximately 50:50 ratio of LMP7 to β5 activity (*Figure 3*), to treatment with ONX 0914. Similar recovery of LMP7 activity following ONX 0914 inhibition was also observed in this cell line (*Figure 5B*) and confirmed in primary human PBMCs from healthy donors (*Figure 5C* and *Figure 5—figure supplement 2*). To establish whether iP recovery is associated with transcriptional upregulation of *PSMB8* (LMP7), qRT-PCR was performed and revealed increased *PSMB8* mRNA levels in the cell lines and the PBMCs with kinetics that correlated with their relative rates of recovery in activity (*Figure 5D*). In particular, the SUP-B15 cell line displayed a rapid spike in *PSMB8* levels before subsequent reduction, presumably due to an elevated rate of translation or *PSMB8* degradation. Immunoblotting was performed and indicated a correlated induction of LMP7 on the protein level in both of the cell lines and in the PBMCs (*Figure 5—figure supplement 3*).

Notably, in the absence of an appreciable change in β5 activity, we also observed a compensatory increase in *PSMB5* mRNA and β5 protein levels (*Figure 5D* and *Figure 5—figure supplement 3*). Inhibition of the cP is well known to induce expression of new cP subunits through the transcription factor Nrf1 upon its proteolytic cleavage and release from the endoplasmic reticulum (ER) membrane as a response to proteasome-induced stress (*Radhakrishnan et al., 2010*; *Sha and Goldberg, 2014*; *Radhakrishnan et al., 2014*; *Weyburne et al., 2017*). Here, an increase in processed Nrf1 was observed following selective iP inhibition (*Figure 5E*) that correlated with the production of *PSMB5*; however, this production was decoupled from an associated change in total β5 activity, indicating a cellular mechanism for maintenance of homeostatic iP and cP activity levels. Together, our results clearly highlight the contribution of the iP to UPS capacity and suggest that immune cells counter a loss in iP load through selective recovery of iP activity, balancing the relative capacities of both proteasomes.

## Discussion

The iP has emerged as a promising therapeutic target for the treatment of autoimmune disorders and cancer; however, it is not well understood how the substrate specificities of the iP and cP diverge. Our global and unbiased peptide library-based strategy (MSP-MS) revealed that the iP and cP have overall similar substrate specificities but that certain cleavage preferences exist for each proteasome. The iP displays a particularly increased preference for cleavage following certain bulky hydrophobic amino acid residues such as tryptophan, whereas the cP displays a greater preference for cleavage following smaller and polar amino acid residues. Indeed, supporting crystallographic evidence suggests that LMP7 has a larger S1 pocket than β5 due to a Met45 conformation that allows for greater access to the binding pocket (*Huber et al., 2012*). Differences may exist in profiling peptide versus protein-based substrates (as well as between biochemical and cellular contexts); however, our peptide library-based approach has a key advantage of affording chemically defined substrates with uniform amino acid distribution. The differences in P1 specificity discovered combined with additional non-prime-side specificity determinants were exploited to develop LMP7- and β5-selective fluorogenic substrates for monitoring the activity of each proteasome under cellular conditions. Compared to activity-based probes, fluorogenic substrates have the advantage of enzymatic signal amplification and enabling a continuous assay readout. In contrast, activity-based probes afford direct subunit labeling for downstream biochemical or cellular applications and are a more direct template for inhibitor development. We envision that the global substrate specificity features identified in this study will aid in next-generation isoform-selective probe and inhibitor design and improve the prediction of endogenous cleavage-site preferences.

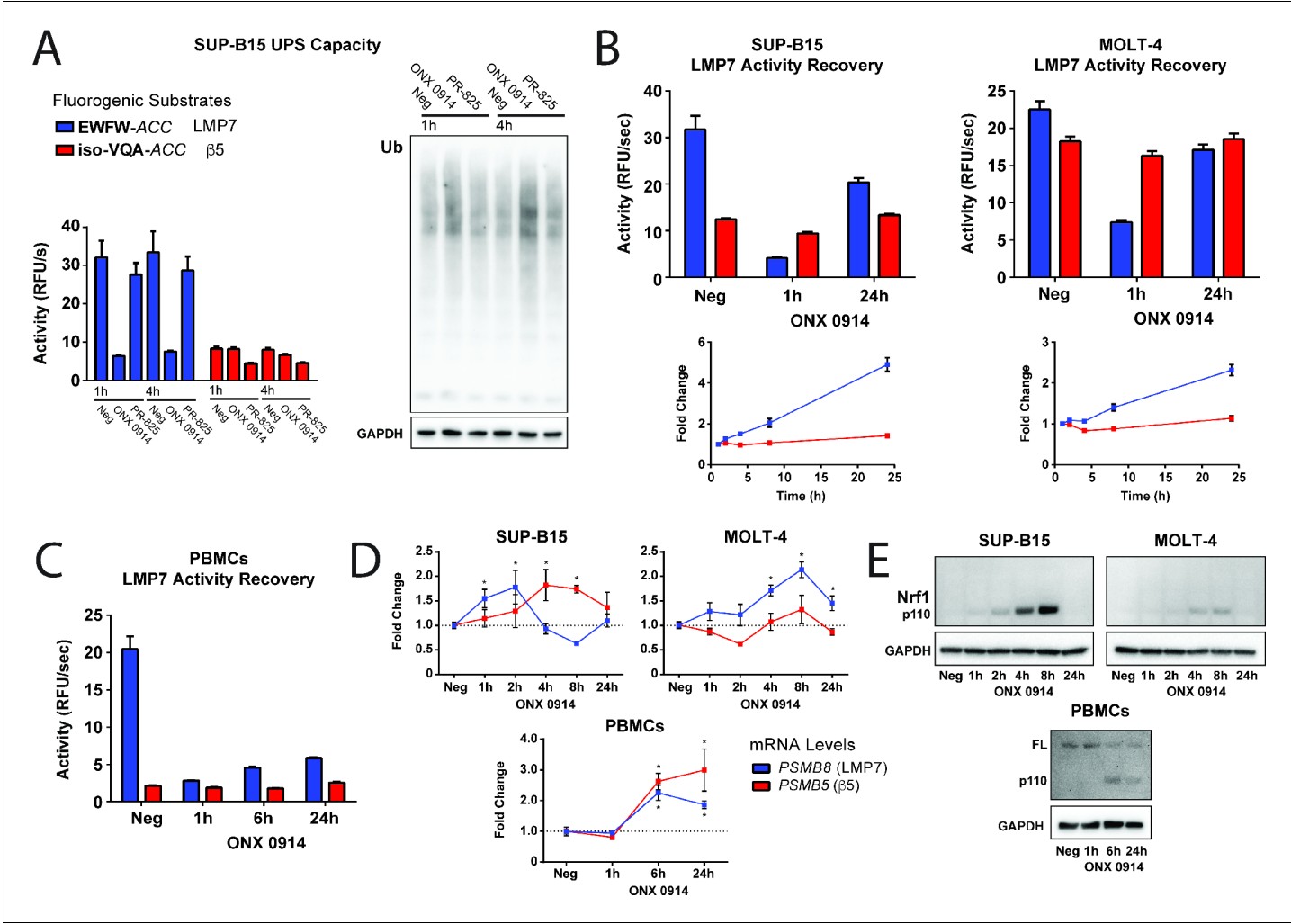

**Figure 5.** iP activity makes a significant contribution to UPS capacity and undergoes selective and coordinated recovery from inhibition. (A) Ubiquitin western blot and matched activity assay for the SUP-B15 cell line (with approximately 80:20 LMP7: β5 activity) showing a substantial increase in global ubiquitination levels upon one-hour pulse treatment with ONX 0914. A significant increase in ubiquitination was not observed with PR-825 treatment, suggesting that LMP7 is a more dominant contributor to UPS capacity in this cell line in accordance with the proportionally higher LMP7 activity level. Selective recovery of LMP7 activity over 24 hr in the (B) SUP-B15 and MOLT-4 cell lines and in (C) primary PBMCs from a healthy donor (see also *Figure 5—figure supplement 2*) upon one-hour pulse treatment with ONX 0914. The SUP-B15 and MOLT-4 lower panels show the fold change in LMP7 and β5 activity levels normalized to the one-hour time point. For all activity assays, mean activity is reported with error bars representing the standard deviation from $n$ = 3–6 replicates. (D) Corresponding qRT-PCR analysis of *PSMB8* (LMP7) and *PSMB5* (β5) mRNA dynamics in the cell lines and PBMCs following ONX 0914 treatment. Mean expression levels are reported with normalization to *GAPDH*. Statistically significant differences are indicated (*$p \leq 0.05$) as determined using a Student's $t$-test with error bars representing the SEM from $n$ = 3 replicates. (E) Corresponding western blots demonstrating accumulation of processed Nrf1 (p110) in RIPA lysates from the cell lines and PBMCs following ONX 0914 treatment. Full-length (FL) Nrf1 was observed in the PBMC lysates. .

DOI: https://doi.org/10.7554/eLife.27364.021

The following figure supplements are available for figure 5:

**Figure supplement 1.** Quantification of the ubiquitin levels in the SUP-B15 cell line at the one- and four-hour time points following one-hour pulse treatment with ONX 0914 and PR-825 (*Figure 5A*).

DOI: https://doi.org/10.7554/eLife.27364.022

**Figure supplement 2.** Recovery of LMP7 activity in PBMCs from $n$ = 3 donors over 24 hr following one-hour pulse treatment with ONX 0914.

DOI: https://doi.org/10.7554/eLife.27364.023

**Figure supplement 3.** Western blots of LMP7 and β5 subunit levels during the ONX 0914 recovery time course in the SUP-B15 and MOLT-4 cell lines and in PBMCs from a healthy donor.

DOI: https://doi.org/10.7554/eLife.27364.024

Degradation of protein substrates by the proteasome generates the pool of peptides made available for display to cytotoxic CD8+ T cells during the MHC I antigen presentation pathway. Proteasomal cleavage preferences are reflected in the C-terminal specificity of MHC I peptides, although additional peptide selection steps further shape the MHC I peptide repertoire, including trafficking into the ER through TAP (transporter associated with antigen processing), binding to allelic-diverse MHC I molecules, and N-terminal trimming by ER-resident aminopeptidases (*Rock et al., 2004*). Initial biochemical studies following the discovery of the iP subunits led to the long-held suggestion that the specificity of the iP is better suited for the generation of peptides for display by MHC I molecules. However, these studies relied on querying limited sets of peptide substrates and inhibitors or on the degradation of model protein substrates. They provided, at times, conflicting evidence that the iP has an increased preference for cleavage following certain basic (arginine), aromatic (phenylalanine and tyrosine), or branched chain amino acids (leucine, valine, and isoleucine) (*Gaczynska et al., 1993*; *Driscoll et al., 1993*; *Gaczynska et al., 1994*; *Cardozo and Kohanski, 1998*; *Toes et al., 2001*; *Boes et al., 1994*; *Eleuteri et al., 1997*). Furthermore, advances in mass spectrometry have only recently made deep coverage of the MHC I peptidome feasible.

To assess the relative impact of iP specificity on the MHC I peptide repertoire, we compared the cP and iP MSP-MS specificity profiles to a large proteomic data set of MHC I-bound peptides. We further queried iP and cP cleavage preferences using a synthetic peptide library that mimics the protein sequences from which these MHC I peptides are derived. This analysis suggested that the majority of MHC I peptides have a similar potential to be generated by either the iP or cP (within a 2-fold difference in cleavage rate). We also show that preferred substrates exist for each proteasome isoform; however, iP specificity differences may primarily alter the extent of display of these epitopes. This appears to only partially result from the increased preference of the iP for specific P1 residues. For example, tryptophan was found to be the most iP-selective P1 amino acid residue but de-enriched at the C-terminal position of displayed MHC I peptides compared to its natural abundance in the human proteome.

Recent evidence suggests that the iP and cP have comparable protein-level selectivity because they bind ubiquitin conjugates similarly and degrade ubiquitinated proteins at similar rates (*Nathan et al., 2013*). In the absence of differences in upstream protein-level selectivity, divergence in iP and cP cleavage preference likely shapes the amount of certain epitopes available for antigen presentation, in combination with basal or conditional (e.g., interferon-induced) differences in proteome composition and the relative activity levels of each proteasome in a given cell. The significant overlap in iP and cP cleavage specificity reported here is consistent with the immune repertoire identified from the splenocytes of iP knockout mice. This repertoire differed from that of wild-type mice by only 50 percent, and furthermore, only certain viral epitopes exhibited an appreciable reduction in display during viral infection in this model (*Kincaid et al., 2011*). These findings have supported the suggestion that the 'quantity' of substrate cleavage, as opposed to strong 'qualitative' differences in specificity, (*Mishto et al., 2014*) impacts the contribution of each proteasome to the MHC I peptide content.

In addition to differential production of certain peptides for antigen presentation, we have shown that iP activity contributes to basal UPS capacity and undergoes a selective recovery response to counter a critical loss in protein load. Irreversible inhibition of LMP7 in immune cells results in selective recovery of LMP7 activity, which is correlated with transcriptional up-regulation of *PSMB8*. However, in immune cells with predominant iP activity, inhibition of LMP7 results in clear compensatory induction of *PSMB5* in the absence of an appreciable change in β5 activity, suggesting that regulation of cP stability, assembly, and/or activation ultimately serves to maintain homeostatic iP and cP activity levels. The transcription factor Nrf1 is well known to mediate production of new cP subunits as a stress recovery response to cP inhibition (*Radhakrishnan et al., 2010*; *Sha and Goldberg, 2014*; *Radhakrishnan et al., 2014*; *Weyburne et al., 2017*). In our study, Nrf1 activation is correlated with production of *PSMB5* following inhibition of the iP. Recovery of the iP under basal conditions appears to occur through a distinct (Nrf1-independent) mechanism; however, basal iP and cP expression are clearly coupled, as an imbalance in iP capacity results in induction of subunits from both proteasomes.

Gene duplication in jawed vertebrates resulted in the emergence of the iP. We have shown that the iP and cP have overlapping substrate specificities as well as distinct cleavage preferences. In addition, both proteasomes are controlled through distinct yet interconnected mechanisms of

regulation. Divergence in regulation balances the relative contribution of each proteasome to basal cellular capacity and in addition, enables induction of the iP during an immune response. Thus, relative cellular capacity and differences in cleavage efficiency for certain peptides likely contribute to iP-specialized roles in signaling and epitope presentation.

## Materials and methods

### Biological reagents

Purified human immunoproteasome core particle (i20S) (cat. E-370), constitutive proteasome core particle (c20S) (cat. E-360), and PA28$\alpha$ (cat. E-380) were purchased from Boston Biochem (Cambridge, MA). Proteasome subunit stoichiometry was assessed as described in the Pro-CISE section below. The following antibodies were purchased from Cell Signaling Technology (Danvers, MA): GAPDH (D16H11) XP rabbit mAb (cat. 5174), ubiquitin (P4D1) mouse mAb (cat. 3936), PSMB8/LMP7 (1A5) mouse mAb (cat. 13726), and PSMB5 (D1H6B) rabbit mAb (cat. 12919). The following HRP-conjugated secondary antibodies were used as appropriate: goat anti-mouse IgG (cat. 172–1011, Bio-Rad, Hercules, CA) and goat anti-rabbit IgG (cat. ab97051, Abcam, Cambridge, MA). TaqMan gene expression assays were purchased from Life Technologies (Waltham, MA) for *PSMB5* (GeneID: 5693, Assay ID: Hs00605652_m1), *PSMB8* (GeneID: 5696, Assay ID: Hs00544758_m1), and *GAPDH* (GeneID: 2597, Assay ID: Hs02786624_g1); all TaqMan assays contained MGB probes with a FAM reporter dye.

### Cell culture

The following human cancer cell lines were purchased from American Type Culture Collection (ATCC, Manassas, VA) – SUP-B15 (cat. CRL-1929) and MOLT-4 (cat. CRL-1582) – and were used directly with minimal passaging. Authentication of cell lines was performed at ATCC using STR profiling, and cell lines were mycoplasma negative. The cell lines used are not contained on the list of commonly misidentified cell lines from the International Cell Line Authentication Committee. Primary human peripheral blood mononuclear cells (PBMCs) (cat. PB005F) were purchased from AllCells (Alameda, CA). The SUP-B15 cell line was propagated in RPMI-1640 medium (2.0 g/L glucose) with 20% FBS, and the MOLT-4 cell line was propagated in RPMI-1640 medium (4.5 g/L glucose) with 10% FBS. Cells were incubated at 37°C under a 5% $CO_2$ atmosphere.

### Multiplex substrate profiling by mass spectrometry (MSP-MS)

Global substrate specificity profiling was performed on PA28$\alpha$-activated 20S proteasomes with the MSP-MS assay using established methods (*O'Donoghue et al., 2012*; *O'Donoghue et al., 2015*; *Winter et al., 2016*). Briefly, human i20S or c20S was activated with 12 eq. of PA28 activator $\alpha$ subunit according to the manufacturer's instructions. Assays were performed using two biological replicates of each proteasome. Proteasomes from different biological replicates were normalized based on activity against the fluorogenic substrate Suc-LLVY-*AMC* (catalog number AS-63892, Anaspec, Fremont, CA ). Activated proteasomes and corresponding PA28$\alpha$-only no-enzyme control were pre-incubated at room temperature for 1 hr in assay buffer (20 mM Tris-HCl, pH 8.0, 0.5 mM EDTA). Following pre-incubation, all samples were diluted two-fold into a 228-member library of peptides pooled in assay buffer (500 nM final peptide concentration). Aliquots (20 µL) were removed after the indicated time points and quenched with 4 µL of 2% formic acid. Samples were desalted using C18 LTS tips (Rainin, Oakland, CA) and rehydrated using 0.2% formic acid prior to mass spectrometry acquisition.

Peptide sequencing was performed on an LTQ Orbitrap-XL mass spectrometer (Thermo) equipped with an EASY-Spray ion source (Thermo) and 10,000 psi nanoACQUITY Ultra Performance Liquid Chromatography system (Waters, Milford, MA). Peptide liquid chromatography was performed on an EASY-Spray PepMap $C_{18}$ column (Thermo, ES800; 3 µm bead size, 75 µm x 150 mm) at a 300 nL/min flow rate from 2% to 50% (vol/vol) acetonitrile in 0.1% formic acid. Survey scans for MS/MS analysis were recorded over a 325–1500 *m/z* mass range. Peptide fragmentation was carried out with collision-induced dissociation (CID) on the six most intense precursor ions, with a minimum of 1000 counts, using an isolation width of 2.0 Th, and a minimum normalized collision energy of 25.

Mass spectrometry peak lists were generated using previously reported software called PAVA (*Guan et al., 2011*). To identify peptide cleavage products, data searches were performed against the library of 228 peptides using Protein Prospector software (http://prospector.ucsf.edu/prospector/mshome.htm, UCSF) (*Chalkley et al., 2008*). Octapeptide (P4-P4') cleavage products used to generate the specificity profiles, the sequences of the 228-member peptide library, and supporting mass spectrometry peptide reports are provided (see *Figure 1—source data 1*). All raw spectrum (.RAW) files from the MSP-MS experiments are available at the massIVE resource (https://massive.ucsd.edu/ProteoSAFe/static/massive.jsp; massIVE accession: MSV000081698). We note that the peptide library was synthesized with norleucine in place of methionine. In database searching, leucine is used to represent norleucine. Tolerances of 20 ppm and 0.8 Da were used for parent and fragment ions, respectively. The following variable modifications were selected with a maximum of 2 modifications per peptide: amino acid oxidation (proline, tryptophan, and tyrosine) and N-terminal pyroglutamate conversion from glutamine. Protein Prospector score thresholds were set to 22 and 21 with maximum expectation values of 0.01 and 0.05 for protein and peptide matches, respectively. Only overlapping peptide cleavages between biological replicates of each proteasome and those not appearing in the no-enzyme control are reported. Peptide cleavage products were imported into iceLogo software v.1.2 to generate protease substrate specificity profiles as described (*Colaert et al., 2009*). Octapeptides (P4-P4') corresponding to the peptide cleavage products were used as the positive dataset, and octapeptides corresponding to all possible cleavage sites in the 228-member library were used as the background dataset (*Figure 1—source data 1*).

## MHC I peptide library cleavage assay

Proteasome cleavage assays against the library of 89 MHC I peptides and subsequent mass spectrometry work-flows were carried out as described above for the MSP-MS assay with the following exceptions. Assays were performed against the MHC I peptide library (500 nM final concentration in assay buffer) using 100 nM c20S or i20S that had been activated with 1200 nM PA28α. Aliquots (10 μL) were removed after 15, 30, 60, 120, 240, 1440, and 2880 min and quenched with 2 μL of 2% formic acid. Assays were performed in duplicate with 2 LC-MS/MS injections per sample (*n* = 4). Liquid chromatography was performed from 2% to 20% (vol/vol) acetonitrile in 0.1% formic acid.

The sequences of the MHC library and supporting mass spectrometry peptide reports are provided (see *Figure 4—source data 1*). All raw spectrum (.RAW) files from the MHC peptide cleavage assay are available at the massIVE resource (https://massive.ucsd.edu/ProteoSAFe/static/massive.jsp; massIVE accession: MSV000081699). Database searching was performed against the library of peptides using Protein Prospector with protein and peptide score thresholds set to 15. MS1 extracted ion chromatograms for label-free quantitation were obtained using Skyline software (v.3.5; University of Washington) (*MacLean et al., 2010*) on quantifiable precursors identified in both independent assays. Initial rates were calculated from a linear fit of the progress curves using Prism (v.6.0) and averaged for calculation of relative cleavage rates. Initial rates were averaged for both cleavage products from a given substrate when possible. Statistical significance (*p*-values) was calculated with a two-tailed Student's *t*-test.

## Positional scanning-synthetic combinatorial library (PS-SCL) profiling

Non-prime-side substrate specificity profiling was performed with a complete, diverse library of 160,000 fluorogenic substrates containing the general structure acetyl-P4-P3-P2-P1-(7-amino-4-carbamoylmethylcoumarin, ACC) (*Harris et al., 2000*). For each sub-library, 10 nM human i20S or c20S underwent SDS pre-activation for 1 hr at room temperature in assay buffer (20 mM Tris, pH 8.0, 0.5 mM EDTA) containing 0.03% SDS. Inhibitor pre-incubations were performed in the presence of 40 nM CFZ. Activity assays were carried out over 1 hr following two-fold sample dilution in the same 0.03% SDS-containing buffer that also contained a 250 μM final concentration of PS-SCL substrate pool. Activity was recorded in black 96-well round bottom plates (Costar, COrning, NY) using a Bio-Tek Synergy H4 Hybrid Multi-Mode Microplate Reader set to $\lambda_{ex}$ = 380 nm and $\lambda_{em}$ = 460 nm (gain = 63). Initial velocity in relative fluorescence units per second (RFU/s) was calculated using a linear fit of the progress curves with Gen5 software v.2.03.

## Synthesis of fluorogenic peptide substrates

Fluorogenic substrates for the iP and cP were prepared using standard solid-phase peptide synthesis protocols. For peptides bearing a C-terminal ACC fluorophore, P1 amino acids were coupled to previously described ACC-substituted Rink amide resin (*Maly et al., 2002*). P1 residues underwent double coupling for two 24 hr periods in *N,N*-dimethylformamide (DMF) with Fmoc-protected P1 amino acid (5 eq.), HATU (5 eq.), and 2,4,6-collidine (5 eq.). For the P4, P3, and P2 positions, double couplings were carried out for two 1 hr periods with Fmoc-amino acid (6.5 eq.), HBTU (6.5 eq.), and N-methylmorpholine (13 eq.). Fmoc deprotection for each step was afforded with 20% 4-methylpiperidine in DMF (v/v). For N-terminal capping groups, double acetylation (Ac) was performed for two 1 hr periods with a 1:1:5:5 (v/v) solution of acetic anhydride (excess), triethylamine (TEA), DMF, and dichloromethane (DCM). N-morpholinyl (m) or 5-methylisoxazolyl (iso) addition was afforded with double coupling for two 1 hr periods with 4-morpholinylacetic acid hydrochloride (6.5 eq.), HBTU (6.5 eq.), and N-methylmorpholine (26 eq.) or 5-methylisoxazole-3-carboxylic acid (6.5 eq.), HBTU (6.5 eq.), and N-methylmorpholine (13 eq.), respectively.

An internally quenched fluorogenic substrate GEWHWPS (P5-P2') was synthesized bearing an N-terminal 7-methoxycoumarin (MCA) fluorophore and C-terminal 2,4-dinitrophenyl (DNP) quencher with the sequence D-Arg-D-Arg-Lys(MCA)-Gly-Glu-Trp-His-Trp-Pro-Ser-Lys(DNP). Residues comprising the GEWHWPS sequence were coupled to preloaded Fmoc-Lys(DNP) Wang resin (AnaSpec) using a Symphony Quartet 4-channel peptide synthesizer (Protein Technologies, Tucson, AZ). Double couplings for all amino acids were carried out in DMF with Fmoc-amino acid (6.5 eq.), HBTU (6.5 eq.), and N-methylmorpholine (13 eq.) as described above with the following exception. The Fmoc-Lys(MCA)-OH (AnaSpec, Inc) fluorophore (3 eq.) underwent a single overnight coupling with HBTU (3 eq.) and N-methylmorpholine (6 eq.).

Trifluoroacetic acid (TFA) cleavage and amino acid side chain deprotection were carried out with a solution (v/v) of TFA (95%), water (2.5%), and triisopropylsilane (2.5%). Peptides were precipitated into diethyl ether, and the crude product was dried under ambient conditions for an overnight period. Peptides were purified by reversed phase high performance liquid chromatography (HPLC) on a Vydac $C_{18}$ column (10 μm bead size, 22 mm x 250 mm) using an Agilent 1200 Series or Waters 2535 preparative HPLC system. Peptide separation was carried out at a flow rate of 10 mL/min using a gradient of 95% acetonitrile in 0.1% aqueous TFA. Peptide purity was assessed by peak area ($A_{330}$) using reversed phase HPLC with an Agilent 1100 Series analytical HPLC system. Peptides were resolved on a Vydac $C_{18}$ column (10 μm bead size, 4.6 mm x 250 mm) at 1 mL/min using a 46 min linear gradient from 5% to 95% acetonitrile in 0.1% aqueous TFA.

Mass spectra were recorded for the following substrates with LC-MS using a Thermo MSQ Plus mass spectrometer in positive ion mode. MS Calcd. (Found): [M + H] $NH_2$-PDFY-ACC 741.3 (741.4); [M + H] $NH_2$-EWHW-ACC 857.3 (857.5); [M + H] $NH_2$-SHRn-ACC 712.3 (712.5); [M + H] $NH_2$-InQT-ACC 674.3 (674.3). Mass spectra were recorded for the following substrates on an Applied Biosystems Voyager DE-STR MALD-TOF in positive ion mode using a 1:1 α-cyano-4-hydroxycinnamic acid: sample ratio (v/v). MS Calcd. (Found): [M + H] $NH_2$-PWHW-*ACC* 825.3 (824.9); [M + H] $NH_2$-PDHW-*ACC* 754.3 (754.1); [M + H] $NH_2$-PDEW-*ACC* 746.3 (746.1); [M + H] $NH_2$-PDFW-*ACC* 764.3 (764.0); [M + H] $NH_2$-KWHW-*ACC* 856.4 (856.0); [M + H] $NH_2$-EKHW-*ACC* 799.3 (799.0); [M + H] $NH_2$-EKFW-*ACC* 809.4 (809.2); [M + H] $NH_2$-EWFW-*ACC* 867.3 (867.1); [M + H] $NH_2$-YVQA-*ACC* 680.3 (680.1); [M + H] $NH_2$-YVQS-*ACC* 696.3 (696.0); [M + H] $NH_2$-YVQG-*ACC* 666.3 (665.9); [M + H] $NH_2$-IVQA-*ACC* 630.3 (630.1); [M + H] $NH_2$-YFQA-*ACC* 728.3 (728.1); [M + H] $NH_2$-YVHA-*ACC* 689.3 (689.1); [M + H] $NH_2$-YVSA-*ACC* 639.3 (639.1); [M + H] $NH_2$-YVIA-*ACC* 665.3 (665.2); [M + Na] Ac-YVQA-*ACC* 744.3 (744.0); [M + H] $NH_2$-VQA-*ACC* 517.2 (516.9); [M + Na] Ac-VQA-*ACC* 581.2 (580.9); [M + H] m-VQA-*ACC* 644.3 (644.0); [M + Na] iso-VQA-*ACC* 648.2 (648.0); [M + H] DArg-DArg-Lys(MCA)-Gly-Glu-Trp-His-Trp-Pro-Ser-Lys(DNP) 1847.8 (1848.5).

## MHC I peptide library design and synthesis

Peptides for the MHC I library were designed to include the C-terminal region of known MHC I peptides (at positions 1–7) and the subsequent parent protein sequence (at positions 8–14) to query proteasome cleavage selectivity between positions 7 and 8 (P1/P1'). Peptide sequences for library construction were selected at random from a proteomic database of MHC I peptides (*Bassani-Sternberg et al., 2015*) (*n* = 22,598) based on their predicted iP or cP cleavage score to not bias by

proteasome cleavage preference. The cleavage score was calculated for each peptide by multiplying the P4-P1 amino acid frequencies for the iP and cP obtained from the MSP-MS assay.

A 95-member MHC I library was synthesized (89 were assayed due to solubility reasons) on 12.5-µmol scale using standard solid-phase peptide synthesis protocols with pre-loaded Fmoc-D-Arg (Pbf)-Wang resin (Anaspec), resulting in 15-mer peptides with a C-terminal D-Arg residue. For the 14-mer sequences, double couplings were performed using Fmoc-amino acid (5 eq.), HCTU (5 eq.), and N-methylmorpholine (20 eq.) with a 96-channel Syro II automated peptide synthesizer (Biotage). TFA cleavage was carried out using a 48-position transfer unit (Biotage, Charlotte, NC). Peptides were purified with reversed phase HPLC on an Agilent Pursuit 5 $C_{18}$ column (5 µm bead size, 150 mm x 21.2 mm) using an Agilent PrepStar 218 series preparative HPLC system under the solvent conditions described above. MS1 spectra were recorded on an AXIMA Performance MALD-TOF/TOF mass spectrometer (Shimadzu, Columbia, MD) in positive ion mode using a 1:1 α-cyano-4-hydroxycinnamic acid: sample ratio (v/v).

## Purified proteasome activity assays

Activity assays with individual fluorogenic substrates were carried out as described above for the PS-SCL assays using final concentrations of 5 nM SDS-activated i20S or c20S and 10 µM substrate except where noted below. The commercial substrates Suc-LLVY-*AMC*, Ac-WLA-*AMC* (Boston Biochem, S-330), and Ac-ANW-*AMC* (Boston Biochem, S-320) were used as supplied. To convert cleaved substrate RFU to free AMC or ACC dye concentration, calibration curves were prepared through total enzymatic hydrolysis of ACC- or AMC-containing peptides that had been quantified through amino acid analysis (Alphalyse, Inc.). Substrate selectivity of PA28α- and SDS-activated 20S proteasomes was compared following 1 hr pre-incubation at room temperature either in the presence of 0.03% SDS or with 12 molar equivalents of PA28α activator in the absence of SDS. For substrate selectivity assays with mixed, purified proteasomes, i20S and c20S underwent individual SDS pre-activation prior to assaying with 5 nM final concentration of pooled 20S. Proteasome ratio assays were performed with 10 µM EWFW-*ACC*, 30 µM iso-VQA-*ACC*, and 30 µM Suc-LLVY-*AMC*. All activity measurements were recorded on the BioTek plate reader described above using $\lambda_{ex}$ = 380 nm and $\lambda_{em}$ = 460 nm (gain = 63) for ACC substrates and Suc-LLVY-*AMC*; 345 nm and $\lambda_{em}$ = 445 nm (gain = 63) for Ac-WLA-*AMC* and Ac-ANW-*AMC*; and $\lambda_{ex}$ = 328 nm and $\lambda_{em}$ = 393 nm (gain = 75) for the GEWHWPS IQ substrate. Michaelis-Menten calculations were performed using a non-linear fit in Prism v.6.0.

## Cell lysate proteasome activity assays

Three lysates containing varied proteasome ratios (whole blood, PBMCs, and the MOLT-4 cell line) were assayed in the absence or presence of proteasome inhibitors as described above. Lysates were generated by washing cells in PBS and lysing the resulting cell pellet in a two-fold greater volume of cold lysis buffer (20 mM Tris, pH 8.0, 5 mM EDTA). Total protein concentration was determined using the BCA Protein Assay Kit (Thermo Scientific Pierce) with bovine serum albumin (BSA) as the protein standard. The cell lysates (1.6 mg/ml for whole blood and 0.4 mg/ml for PBMCs and MOLT-4) underwent pre-incubation for 1 hr at room temperature with 100 nM inhibitor (CFZ, ONX 0914, or PR-825) in assay buffer (20 mM Tris, pH 8.0, 0.5 mM EDTA) without SDS. For $IC_{50}$ measurements with the MOLT-4 lysate, pre-incubations were performed with seven inhibitor concentrations ranging from 40 µM to 0.04 nM CFZ, ONX 0914, or PR-825. Activity assays were carried out over 1 hr following two-fold sample dilution in the same assay buffer containing final concentrations of 10 µM $NH_2$-EWFW-ACC, 30 µM iso-VQA-ACC, or 30 µM Suc-LLVY-AMC. Activity measurements were recorded on the BioTek plate reader described above in 384-well plates using either $\lambda_{ex}$ = 380 nm and $\lambda_{em}$ = 460 nm (gain = 85) for the ACC substrates or $\lambda_{ex}$ = 345 nm and $\lambda_{em}$ = 445 nm (gain = 85) for the AMC substrate.

## ProCISE quantification of proteasome subunit levels and $IC_{50}$ values

The ProCISE assay was performed as previously described (*Parlati et al., 2009*) to determine the quantities of chymotrypsin-like subunits in each lysate type. For $IC_{50}$ determinations, diluted lysates (20 mM Tris, pH 8.0, 0.5 mM EDTA) were pretreated with CFZ, ONX 0914, or PR-825 for 1 hr at room temperature before the addition of the proteasome active-site probe (PABP, Nanosyn).

Following 2 hr room temperature incubation with biotinylated PABP, lysate was denatured in guanidine hydrochloride and probe-bound subunits were isolated via incubation with streptavidin-conjugated sepharose beads (GE Healthcare Bio-Sciences, Pittsburgh, PA) in 0.65 µm filter plates (EMD Millipore, Burlington, MA). Samples were rinsed of denaturant and probed overnight at 4°C with subunit-specific primary antibodies. Subsequent to primary antibody removal, HRP-conjugated secondary antibodies (Jackson ImmunoResearch, West Grove, PA) were allowed to bind for 2 hr at room temperature before removal and signal generation utilizing a chemiluminescent substrate (Thermo Scientific Pierce). For subunit level determination in lysates, absolute values of subunit per microgram of total protein were based on a purified proteasome standard curve. For $IC_{50}$ determination, percent activity relative to DMSO-treated controls was calculated for dose-response curves.

In a commercial preparation of c20S (E-360) obtained from Boston Biochem, the subunit levels were found to be as follows by LC-MS/MS analysis (quantification by the emPAI method): β5/LMP7 (92% and 8%), β2/MCEL-1 (92% and 8%), and β1/LMP2 (98% and 2%). For a commercial preparation of i20S (E-370), the subunit levels were found to be as follows: β5/LMP7 (20% and 80%), β2/MCEL-1 (10% and 90%), and β1/LMP2 (48% and 52%). We note that the β1/LMP2 (caspase-like) subunits are the smallest contributor to the overall specificity of both proteasomes (*Figure 1A*).

## Cellular proteasome inhibition and recovery assays

Cellular inhibitor treatments were performed with ONX 0914 (200 nM) and PR-825 (100 or 125 nM) to achieve optimal LMP7 or β5 selectivity, respectively (*Muchamuel et al., 2009*). SUP-B15 and MOLT-4 cells were seeded at $2 \times 10^6$/mL in RPMI media containing 10% FBS 24 hr prior to inhibition assays; PBMC vials were thawed and seeded at $4.4 \times 10^6$/mL in RPMI media with 10% FBS 3 hr prior to inhibition assays. Cells were diluted 2-fold with serum-free RPMI media containing 2X inhibitor or vehicle control (0.5% DMSO and 5% FBS final). Inhibitor treatments were carried out for 1 hr at 37°C, and to remove unbound inhibitor, cells were immediately washed three times (SUP-B15 and PBMCs) or six times (MOLT-4) with D-PBS for 1 hr time points or RPMI media containing 5% FBS for subsequent time points. Cells were incubated at 37°C for the duration of the time course and washed two times in D-PBS to remove serum upon harvest. The cell pellets were flash-frozen in liquid $N_2$ prior to storage at −80°C.

Cell pellets were thawed and incubated for 30 min on ice in non-denaturing lysis buffer (20 mM Tris, pH 8.0, 5 mM EDTA) or RIPA buffer that contained cOmplete™ EDTA-free inhibitor cocktail (Roche, Pleasanton, CA). Total protein concentration in supernatants was determined using a Bradford Protein Assay Kit (Thermo Scientific Pierce) or BCA Protein Assay Kit (Cell Signaling Technology), respectively, with BSA as the protein standard. Lysate activity assays were performed following 2-fold dilution of 0.1–0.4 mg/mL lysate into assay buffer as described above. Corresponding western blots were performed following SDS-PAGE in MOPS buffer with NuPAGE™4–12% Bis-Tris gels (Invitrogen). Wet transfer to Immun-Blot PVDF membrane (Bio-Rad) was performed at 200V and 220A (constant) for 60–90 min. Blocking was performed in 5% milk (in PBS-T), and the antibodies used are described above. Secondary detection was afforded with SuperSignal West Pico or Femto chemiluminescent substrate (Thermo Scientific Pierce). Chemiluminescent imaging was performed with a ChemiDoc™ MP Imaging System (Bio-Rad).

## qRT-PCR.

Total RNA was extracted from cells using the RNeasy Plus Mini kit (Qiagen, Germantown, MD). The quantity and quality of RNA was confirmed by absorption measurements at 260 and 280 nm. Single strand cDNA synthesis and PCR amplification were carried out in a 1-step reaction using the Brilliant II QRT-PCR Master Mix (Agilent, Santa Clara, CA). 100 ng of RNA was loaded in 96-well plate format and amplified with 0.9 µM of primers, 0.25 µM of MGB probe, and 30 nM of ROX passive reference dye. The reaction was carried out in a Mx3005P instrument (Agilent) with the following parameters: an RT step at 50°C for 30 min was followed by a preincubation step at 95°C for 10 min, 50 cycles of denaturation at 95°C for 15 s, and annealing/extension at 60°C for 1 min. Efficiencies for each primer/probe set were calculated from a serial dilution of Human Reference Total RNA (Agilent) and were >95%. Relative quantification of gene expression was calculated with the comparative cycle threshold method, normalizing for *GAPDH* expression levels.

## Proteasome evolution

β5 and LMP7 orthologs were selected from multiple species as defined previously (*Sutoh et al., 2012*). Protein sequences were aligned using MSAProbs (*Liu et al., 2010a*) for each isoform separately. These alignments were then used to align the respective nucleotide sequences of the coding regions for each gene using an in-house python script. We then calculated phylogenetic trees using MrBayes (*Huelsenbeck and Ronquist, 2001*) with default parameter values. Finally, the evolution of sites in the isoform sequences was estimated using the SLR algorithm (*Massingham and Goldman, 2005*).

## Statistical methods

Statistical significance (*p*-values) was determined using a Student's *t*-test. For multiple comparison correction (*q*-values), we used the methodology by Storey (*Storey, 2002*). We chose an FDR level of 0.05 and $\lambda = 0$ to be conservative and also equivalent with the methodology of Benjamini and Hochberg (*Benjamini and Hochberg, 1995*) at $\alpha = 0.05$. Errors bars represent the standard deviation unless otherwise noted. Numbers of replicates associated with each figure are provided in the corresponding figure legend.

## Acknowledgements

Mass spectrometry was performed in collaboration with the UCSF Mass Spectrometry Facility (directed by Prof. Alma Burlingame). This work was supported by the UCSF and Onyx Oncology Innovation Alliance, National Institutes of Health (NIH) grants R21CA186077 (to AS and CSC), R21AI133393 (to AJO), P41CA196276 (to CSC), and P41GM103481 (UCSF Mass Spectrometry Facility), a UCSF Program for Breakthrough Biomedical Research (PBBR) award (to CSC, MBW, and MR), and a UCSF Enabling Technologies Advisory Committee (ETAC) award (to CSC, MBW, and MR). M. B.W. was supported by NIH postdoctoral fellowship F32CA168150. We gratefully acknowledge Dr. Cammie Edwards for program management of the UCSF-Onyx alliance. We also acknowledge Prof. Ryan Hernandez, Dr. Giselle Knudsen, Dr. Judd Hultquist, and Prof. Jason Gestwicki (UCSF) and Dr. Christopher Kirk (Kezar Life Sciences) for helpful discussions.

## Additional information

### Competing interests

Shirin Arastu-Kapur, Tonia J Buchholz, Janet L Anderl: Employee of Onyx Pharmaceuticals, Inc., an Amgen subsidiary. The other authors declare that no competing interests exist.

### Funding

| Funder | Grant reference number | Author |
|---|---|---|
| National Institutes of Health | F32CA168150 | Michael B Winter |
| University of California, San Francisco | Program for Breakthrough Biomedical Research | Michael B Winter<br>Matthew Ravalin<br>Charles S Craik |
| University of California, San Francisco | Enabling Technologies Advisory Committee | Michael B Winter<br>Matthew Ravalin<br>Charles S Craik |
| National Institutes of Health | R21CA186077 | Andrej Sali<br>Charles S Craik |
| National Institutes of Health | R21AI133393 | Anthony J O'Donoghue |
| National Institutes of Health | P41CA196276 | Charles S Craik |

The funders had no role in study design, data collection and interpretation, or the decision to submit the work for publication.

## Author contributions

Michael B Winter, Conceptualization, Formal analysis, Funding acquisition, Investigation, Methodology, Writing—original draft, Writing—review and editing; Florencia La Greca, Formal analysis, Investigation, Methodology, Writing—original draft, Writing—review and editing; Shirin Arastu-Kapur, Conceptualization, Formal analysis, Supervision, Funding acquisition, Investigation, Methodology, Project administration, Writing—review and editing; Francesco Caiazza, Janet L Anderl, Formal analysis, Investigation, Methodology, Writing—review and editing; Peter Cimermancic, Conceptualization, Data curation, Formal analysis, Investigation, Methodology, Writing—review and editing; Tonia J Buchholz, Conceptualization, Formal analysis, Supervision, Investigation, Methodology, Project administration, Writing—review and editing; Matthew Ravalin, Markus F Bohn, Conceptualization, Formal analysis, Investigation, Methodology, Writing—review and editing; Andrej Sali, Conceptualization, Resources, Supervision, Funding acquisition, Methodology, Project administration, Writing—review and editing; Anthony J O'Donoghue, Conceptualization, Formal analysis, Supervision, Funding acquisition, Investigation, Methodology, Writing—original draft, Project administration, Writing—review and editing; Charles S Craik, Conceptualization, Resources, Supervision, Funding acquisition, Project administration, Writing—review and editing

## Author ORCIDs

Michael B Winter http://orcid.org/0000-0003-4824-8024
Francesco Caiazza http://orcid.org/0000-0003-4228-1130
Andrej Sali http://orcid.org/0000-0003-0435-6197
Anthony J O'Donoghue http://orcid.org/0000-0001-5695-0409
Charles S Craik http://orcid.org/0000-0001-7704-9185

## Decision letter and Author response

Decision letter https://doi.org/10.7554/eLife.27364.029
Author response https://doi.org/10.7554/eLife.27364.030

# Additional files

## Supplementary files

• Supplementary File 1. Comparison of the reproducibility of the MSP-MS assay for technical triplicate samples. Percent identity was compared across replicates for the intact library ('no enzyme control') and after the addition of the immunoproteasome as an example. The 8 hr time point was chosen to maximize the number of cleavages in the comparison.
DOI: https://doi.org/10.7554/eLife.27364.025

• Supplementary File 2 Comparison of select ACC substrate Michaelis-Menten parameters.
DOI: https://doi.org/10.7554/eLife.27364.026

• Supplementary File 3. Comparison of compound $IC_{50}$ values in the MOLT-4 lysate obtained using the optimal fluorogenic substrates and the ProCISE assay.
DOI: https://doi.org/10.7554/eLife.27364.027

• Transparent reporting form
DOI: https://doi.org/10.7554/eLife.27364.028

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
