## [Decision Letter]

Thank you for submitting your article "Immunoproteasome Functions Explained by Divergence in Cleavage Specificity and Regulation" for consideration by *eLife*. Your article has been reviewed by two peer reviewers (reviewers 2 and 3), and the evaluation has been overseen by a Guest Reviewing Editor and Michael Marletta as the Senior Editor. The following individuals involved in review of your submission have agreed to reveal their identity: Peter M Kloetzel (Reviewer #1); Herman Overkleeft (Reviewer #2).

The reviewers have discussed the reviews with one another and the Guest Reviewing Editor has drafted this decision to help you prepare a revised submission.

The manuscript is in principle within the scope of *eLife*'s interest/mission. Both reviewers raise valid points, so rather than try to condense these two reviews, I suggest that the authors be given access to the full reviews and be given the opportunity to submit a properly revised version within the time line customarily allotted by *eLife*. I look forward to seeing a revised manuscript.

*Reviewer #1:*

Winter et al., have analysed the differences in cleavage properties of standard and immunoproteasome and they rightly state that these differences still are not fully understood. It has been previously shown that both isoforms in principle have the ability to cleave behind any of the 20 aa residues but that sP and iP depending on the substrate reveal differences in relative cleavage site usage. Based on peptide library based profiling Winter et al. now conclude that sP and iP exhibit overlapping but distinct substrate specificities. The authors also designed novel isoform selective substrates and confirm previous studies by showing that iP function affects proteostasis in cells predominantly expressing iP.

This reviewer appreciates this multi-disciplinary study to improve our knowledge and to eventually confute former models. The paper however raises a number of concerns with regard to the experiments and the conclusions drawn as will be outlined in detail below.

Figure 1.

The authors use a rationally designed peptide library comprising 228 14mer peptides to test their degradation by sP or iP. However, this multi-substrate approach and the data deduced raise a number of question and important information is lacking.

In order to be able to reproduce the experiments one has to know:a) what are the sequences of the peptides in the library?b) what was the rationale behind the design of the peptide library?

Given that the proteasomes can cleave behind any given residue the percentages of cleavage sites identified (18.4% and 21.3% of 2964 potential cleavage sites) seems to be very low and questions the overall sensitivity of the assays. It is well known that different substrates will be degraded with different efficiencies by either sP or iP. However, such differences will strongly influence the validation of cleavage site usage unless the cleavage site usage is not corrected for max/2 turnover of the substrate. Importantly, proteasomes are known to cut less efficiently and also slowly within short peptides. Dolenc et al. (FEBS Lett 1998) identified a length of 13 residues as lower limit to have an efficient degradation rate. Since the MSP-MS method uses 14mers, this reviewer thinks that it is mandatory to repeat the MSP-MS assay with much longer kinetics than 60 min (2,4,6,8 hours).

The authors neither report on the number of experiments performed to generate the data shown in Figure 1 nor on the reproducibility or the overlap between different experiments.

Assuming that technical replicates of the MS analysis and biological replicates (proteasome digestions) were performed, it will be very important to know the number of unique cleavage sites between:

- technical replicates;

- biological replicates;

- biological replicates performed at different time points using either sP or iP. The latter is a particular relevant control. I suppose that the alleged unique cleavage sites reported in Figure 1 will not be confirmed by performing the assays with longer time-point. (Also see comment to Figure 4).

I am also intrigued by the fact that the authors identified 217 cleavage sites being specific for sP. However, as stated in Materials and methods the commercial PBMC derived iP contains between 10% and 48% standard subunits. How can it be that such mixed-type proteasomes cannot cleave at all at sites used by the sP?

Performing and showing the outcomes of the requested controls will help to understand whether sP and iP really have such a large number of unique cleavage sites (348 of 762 cleavage sites identified). Using longer kinetics will most likely dramatically reduce this number. Even in case that there are only very few differences this would not affect the relevance of the study but will modify one of the main take home messages of the study.

I would also suggest to add a supplementary figure comparing the outcome of Figure 1, with those of Toes et al., JEM 2001 and Mishto et al., EurJI 2014 to better understand differences and similarities.

Figure 2.

Figure 2 may serve the purpose of the idea but are not very informative. It is not clear how the experiment was performed. Were these peptides individually processed by either sP or iP and if so, what were the conditions. Are these peptides equally well degraded by both isoforms; are these all the cleavage sites detected by MS?

Figure 2—figure supplement 1.

There seems to be an editorial error for the km value of EWFW-ACC for i20S. It probably should read 1.8 μm instead 7.0 uM.

Figure 3.

It is somewhat surprising that independent of the sP/iP ratio the activity measured with the commonly used substrate Suc-LLVY-AMC remained unchanged. This certainly stands in contrast to several reports by Gaczynska and coworkers (cited) as well as by many others.

Figure 4.

I do not really understand why the authors state that iP clearly expands the MHC-I repertoire when the authors don´t find proteasome isoform specific cleavage sites which generate the tested epitopes. As documented in Figure 4 the most sP preferred epitope HTQVIEL>ERKFSHQ is also generated by iP, albeit less efficiently. Similarly, the most iP favoured epitope is also generated by sP. My interpretation of the data shown is that there are epitopes which are more efficiently generated by either sP or iP which consequently will affect their presentation at the cell surface. (in this context see Zanker, D., Waithman, J., Yewdell, J. W. and Chen, W.. J. Immunol. 2013. 191: 52-59.)

In part the experiments shown also answer my concerns expressed for the data in Figure 1. Here, in Figure 4 the authors have performed longer kinetics as requested above and conclude that many peptides have an equivalent potential of being generated by either proteasome isoform. Even the tryptophan-cleavage site, although preferred by iP is shown to be used by sP as well, albeit less efficiently (Figure 4).

This reviewer has some concerns regarding the label-free quantification of the peptides. It is well known that the MS ion signal strongly varies between peptides of different sequence. This is irrelevant with a dataset of thousands of peptides but render not applicable the label-free strategy to quantify a small number of short peptides (Bassani-Sternberg et al., MCP 2015; Liepe et al., Science 2016; Tenzer et al., Nat Immunol 2009).

In Materials and methods it is stated that 89 epitope derived peptides were analysed but only slightly more the 50 are documented in Figure 5 (please explain).

Figure 5.

The authors show that in cells mostly expressing iP, selective inhibition of LMP7 results in an accumulation of ub-conjugates. They infer from these results that iP activity makes a significant contribution to UPS capacity. This conclusion is in agreement with previously published data Seifert et al. Cell (2010), which I would suggest to cite in this context. That inhibition of the beta5 site doesn´t evoke similar effects is most likely due to the small amounts of sP present in these samples.

The authors observe a bounce back of LMP7 activity after irreversible inhibition of LMP7 as well as upregulation of both PSMB8 and PSMB5 mRNAs during the recovery phase. This is a well-known phenomenon. That PSMB5 mRNAs has little impact on β 5 activity is also to be expected because in the presence of both β 5 and LMP7 units, incorporation of LMP7 is dominant due its higher binding affinity for the proteasome maturation protein POMP.

*Reviewer #2:*

As I see it this paper consists of two parts that are interconnected by the use of the fluorogenic beta5c/beta5i substrates developed in Figure 1–Figure 3 to link iCP activity, set off against cCP activity, in global proteasomal proteolysis (Figure 5) and MHCI ligand production (Figure 4). These two (or three) sections feel somewhat disconnected to me and the main aim: to establish substrate preferences of the various proteasomal activities and to connect these to cellular processes has not been reached to satisfaction. Despite this, the workflow to identify substrate preferences and to derive two new and considerably beta5c and beta5i-selective substrates – the part of the work I am able to judge best – is executed very well and the resulting reagents should be of considerable interest to the research community. I have a few points that may need clarification:

Different proteasome preparations are being used in the various assays: sometimes SDS-activated and sometimes PA28-activated. To what extent are these methods providing similarly activated proteasomes?

Peptides, rather than (unfolded) proteins are being used to establish cleavage preferences. To what extent are peptides suitable protein substitutes? what about concentration, reaction times, other factors that may be different in cells?

Why only beta5c/beta5i?

Are ONX0914 and PR825 – developed for murine proteasomes, equally selective against their respective human proteasome targets (given that recent literature point towards differences in the active sites of especially the chymotryptic sites from these two species)?

Why, in designing selective fluorogenic substrates, not considering non-proteinogenic amino acids, also in view of the literature on likewise-designed inhibitors?

In using the substrates in mixtures containing both proteasome species: how to correct for minor processing of the 'unwanted' proteasome (since the reagents are not truly selective?)

Next top cP and iP species mixed proteasomes exist and that may have other substrate preferences: how does the work take the possible existence of such species into account?

Presentation of the work, I would favour some chemical structures, possibly also a chemical scheme (even though the methodology to synthesise the compounds has been published from the same group some time ago). In general I find the manuscript hard to get into, with lots of abbreviations.

A survey of the pros and cons of fluorogenic substrates in comparison with other methodology (activity-based protein profiling, the Procise assay) would be helpful.

[Editors' note: further revisions were requested prior to acceptance, as described below.]

Thank you for resubmitting your work entitled "Immunoproteasome Functions Explained by Divergence in Cleavage Specificity and Regulation" for further consideration at *eLife*. Your revised article has been favorably evaluated by Michael Marletta as the Senior and Reviewing Editor, and one reviewer.

The manuscript has been improved but there are some remaining issues that need to be addressed before acceptance, as outlined below.

We try hard to avoid requiring more experiments after one round of revision. I ask that you consider the points raised below and send a response to me before initiating a new round experiments (the replicates referred to below).

*Reviewer #1:*

I appreciate the additional work by Winter et al. Their efforts certainly improved the manuscript. In particular, the kinetics experiments and the biological replicates of the MSP-MS assays give now a clearer picture of the results and the potency of the method.

Having said this, there still remain some points which in my mind cast doubt onto the robustness of some the experiments.

The authors show that:a) they have a 70% reproducibility of technical replicate at 8 h digestion (they show it only for immunoproteasome and at 8h, though);b) they have around 70% reproducibility of biological replicate at 8 h digestion (as far as I get from Figure 1—figure supplement 1);c) there is a strong increase of the number of cleavage-sites identified over time;d) they have 19.5% cleavage sites identified only in cP digestions although their iP has 10-48% cP subunits. How can it be that a cP-iP mixed population doesn't use all cleavage-sites used by cP?e) After 8 h, 11-13% of the cleavage-sites have been used by proteasomes.

Comment: The surprisingly low number of cleavage sites identified is not compatible with the extent of cleavage site usage published in the literature and also the fact that proteasomes cleave more or less behind any amino acid. When single polypeptide substrates are analysed a cleavage site coverage of >60% is the rule.

I raised the issue of the sensitivity of the MSP-MS before. In the point by point review this issue is however only partially addressed. I now see the rationale behind the design of the library, but considering the cleavage site coverage in the experiments I start to wonder whether what works perfectly for conventional proteases also will work for the proteasome with extremely little cleavage site specificity. (This does not imply that proteasomes don´t have cleavage site preferences).

The authors propose that the MSP-MS method represents an improvement for understanding the proteasome dynamics. Despite all the work published in the past two decades, understanding proteasome dynamics still is an important issue. Because of this, the authors should try to give a robust answer to the question they raised: does iP differ from cP in terms of cleavage-site usage? Is the truth close to "only quantitative differences" as stated by a few or close to "strong qualitative differences" as published by others?

The authors rephrased several parts of the text to satisfy both hypotheses. They would have, however, the opportunity to provide strong evidence supporting one of the two hypotheses. Such an answer would be a significant improvement in the field and certainly worth publishing.

To this end, the authors ought to strengthen their MS analysis. Here I have some suggestions:

a) The authors should perform more technical replicates than 2 (5 are sufficient in our set up), considering the reproducibility that they have.

b) They should add the new peptides identified in each replicate to the final list of identified peptides (and thus cleavage-sites used), instead of focusing only on the common peptide products. Because if only the common peptide products (and cleavage-sites) are true, it means that the authors have an experimental FDR =30%. Clearly too high to say anything about differences between cP and iP.

For example, in Figure 1—figure supplement.1B I see 101 cleavage-sites used by cP and 134 by iP after 1h. In the first submitted Figure 1, they had 631 cleavage-sites for cP and 545 for iP. What did happen to the others? I guess the discrepancy is due to the new data analysis used. In fact, even considering the cleavage-sites identified in only one of the biological replicates, in the new Figure 1—figure supplement 1, after 1 h digestion, I see around 220 cleavage-sites identified in cP and 250 in iP.

I suggest, that the authors should try to increase the coverage of the peptide products. Considering only the products common to the replicates the authors likely analyse only the most abundant peptide products. Therefore, my question is, do the authors want to study only the most frequently used cleavage-sites (top 10 I would say since they identified only 11-13% of the cleavage-sites) or all cleavage-sites used by proteasome? If the intention is to focus on the top ten only, they should re-phrase the entire manuscript.

With 5 technical replicates, they could have a better view of the cleavage-site usages by proteasome isoforms. And I guess they will also increase the number of cleavage-sites used by proteasome since 11-13% is really not compatible with proteasomes cleaving after almost any amino acid (see also above).

c) The authors may perform longer digestions. In their experiments, the authors see a decline in the number of newly identified cleavage-sites over time. However, this could also be due to inhibitory effects of the products on proteasome activity (see Liepe et al., *eLife* 2015). To minimize this problem, the authors could (should) also increase the concentration of proteasome.

Everything that it is useful to identify at least 40-50% of the potential cleavage sites or at least reaching a plateau in terms of cleavage-sites used by proteasome. Otherwise the authors are watching only the tip of the iceberg.

d) Concerning the MS analysis, FDR=0.05 is not conservative. The other parameters (Materials and methods) should be better clarified without referring to other studies.

I wonder if the analysis of the MHC-I peptides using the outcome of the MSP-MS would be different using a more sensitive approach and if it would show differences between cP and iP.

e) The authors added the statement: "Proteasomes from different biological replicates were normalized based on activity against the fluorogenic substrate Suc-LLVY-AMC (Anaspec, AS-63892)".

It is generally accepted that the hydrolytic activity measured with short fluorogenic substrates such as Suc-LLVY-AMC may have very little to do with the ability of proteasomes to degrade larger polypeptide substrates. Identical peptide hydrolysing activities or adjustment of the amounts of proteasomes to obtain identical activities is by no way a reliable method if one wants to study the processing of polypeptides by different proteasome subtypes or by different commercially available proteasome charges.

As pointed out in my first review in a different context, it is essential to monitor the turnover of the substrate(s) and compare fragment patterns or cleavage site usage when approx.. 40% of the substrates is turned over. In this way, it is possible to compare the cleavage site usage of two proteasome samples more or less independent of their activity and independent of different substrate degradation kinetics (which often differ between substrates and certainly will do so in a library).

f) The authors should make the RAW files of the XL measurements available by depositing them in an online archive. They should also provide in Figure 1—source data 1 and Figure 4—source data 1 enough info (MS scan, RT) to reproduce the results. They also ought to provide the list of the peptide library (not on a collaboration-base) as supplementary material in this manuscript. I think this is the *eLife* policy. The aim is allowing other scientists to reproduce the results

g) The Supplementary file 1 is not clear. I did not find a caption explaining the meaning of #1, #2, #3. Are they biological replicates? No enzyme is the control without any enzyme? If yes, where are the peptides listed identified in the No Enzyme Control? Why is the reproducibility of the cP not reported here?

---

## [Author Response]

Reviewer #1:[…] This reviewer appreciates this multi-disciplinary study to improve our knowledge and to eventually confute former models. The paper however raises a number of concerns with regard to the experiments and the conclusions drawn as will be outlined in detail below.Figure 1.The authors use a rationally designed peptide library comprising 228 14mer peptides to test their degradation by sP or iP. However, this multi-substrate approach and the data deduced raise a number of question and important information is lacking.In order to be able to reproduce the experiments one has to know:a) what are the sequences of the peptides in the library?b) what was the rationale behind the design of the peptide library?

We appreciate the necessity for any group to reproduce the experiments outlined in the study. To address this point, we have made the peptide library available through academic collaboration with our research groups (subsection “Multiplex Substrate Profiling by Mass Spectrometry (MSP-MS)”, last paragraph). In Figure 1—source data 1, we have provided a list of all iP and cP cleavage site data used to generate Figure 1 and 2.

The MSP-MS peptide library has been described in O’Donoghue et al. Nat. Methods 2012 (PMID:23023596). The library design is based on the specificity preference of proteases for two optimally positioned amino acids within a substrate. For example, the amino acids in the P1 and P1ʹ position generally drive the specificity of aspartic acid proteases, whereas matrix metalloproteases have important substrate binding interactions in the S2 and S1ʹ sub-sites. In another example, the substrate specificity for caspase 3 and granzyme B is primarily located in the S4 and S1 pockets. Therefore, we designed an unbiased and diverse library of 14-mer peptides containing all possible neighbor (XY) and near neighbor (X*Y and X**Y) amino acid pairs so that all proteases would be able to cleave a selection of peptides. We have shown that highly specific proteases, such as human rhinovirus-14 protease, cleave a single peptide within the library (Nat. Methods 2012), whereas proteases with broad specificity, such as human cathepsin E, cleave 277 peptide bonds (Ivry et al. Clin Cancer Res 2017; PMID: 28424202). The mathematical formula used to design these sequences was published in our 2012 Nat. Methods paper.

Given that the proteasomes can cleave behind any given residue the percentages of cleavage sites identified (18.4% and 21.3% of 2964 potential cleavage sites) seems to be very low and questions the overall sensitivity of the assays. It is well known that different substrates will be degraded with different efficiencies by either sP or iP. However, such differences will strongly influence the validation of cleavage site usage unless the cleavage site usage is not corrected for max/2 turnover of the substrate. Importantly, proteasomes are known to cut less efficiently and also slowly within short peptides. Dolenc et al. (FEBS Lett 1998) identified a length of 13 residues as lower limit to have an efficient degradation rate. Since the MSP-MS method uses 14mers, this reviewer thinks that it is mandatory to repeat the MSP-MS assay with much longer kinetics than 60 min (2,4,6,8 hours).

In the Dolenc et al.paper, the cleavage rate of a 14-mer peptide by *Thermoplasma acidophilum* proteasome is equivalent to a 16-mer and 23-mer peptide. Therefore, the length of our starting substrates is likely to be sufficient for these assays. However, to address the concern from the reviewer about cleavage events that take place after 1 h, we have repeated the MSP-MS assay and performed an 8-hour time course. These additional time points have allowed for the generation of progress curves for iP and cP cleavage of the library (Figure 1—figure supplement 1). As predicted by the reviewer, the number of new cleavage sites identified increases with time; however, the rate of new cleavage sites generated by each proteasome reduces as the assay progresses (Figure 1—figure supplement 1). We have also provided heat maps to compare proteasome specificity throughout the extended assay time course (Figure 2—figure supplement 2).

The authors neither report on the number of experiments performed to generate the data shown in Figure 1 nor on the reproducibility or the overlap between different experiments.Assuming that technical replicates of the MS analysis and biological replicates (proteasome digestions) were performed, it will be very important to know the number of unique cleavage sites between:- technical replicates;- biological replicates;- biological replicates performed at different time points using either sP or iP. The latter is a particular relevant control. I suppose that the alleged unique cleavage sites reported in Figure 1 will not be confirmed by performing the assays with longer time-point. (Also see comment to Figure 4).

All MSP-MS assays have now been performed with two commercial stocks of cP and iP (different lot numbers). In our manuscript, only cleavage sites that were found in two independent cP assays and in two independent iP assays (summarized in Figure 1—figure supplement 1) are reported and discussed. We have modified the main text (subsection “Global Substrate Specificity Profiling of the iP and cP”, first paragraph) and figure legends to clearly state that only cleavage sites that are common between two biological replicates are considered in our study.

For technical replicates, we have shown that the number of peptides identified in duplicate technical replicate assays ranges from an average of 78.5% at the start of the assay (before addition of proteasome) to 70.0% after 8 hours of incubation with proteasome. A table showing these data has been included in Supplementary file 1.

As suggested, we have provided the overlap between biological replicates throughout the assay time course (Figure 1—figure supplement 1). This analysis indicated that indeed the iP and cP have increased cleavage overlap at longer time points but that the proteasomes also retain unique specificity features (Figure 1—figure supplement 2). We feel that this experiment helped strengthen the message of the manuscript and thank the reviewer for the suggestion.

I am also intrigued by the fact that the authors identified 217 cleavage sites being specific for sP. However, as stated in Materials and methods the commercial PBMC derived iP contains between 10% and 48% standard subunits. How can it be that such mixed-type proteasomes cannot cleave at all at sites used by the sP?

The reviewer has correctly pointed out that each proteasome sample used in this study has β1, β2, β5, LMP2, MCEL-1 and LMP7 subunits although the ratio of each subunit differs considerably between the iP and cP preparations. With extended incubation time, iP and cP may eventually hydrolyze the peptide library into the same set of cleavage products. The differences in cleavage site preferences that are detected in our assay over the course of 8 hours are likely due to differences in cleavage efficiency. Therefore, we have removed the term “unique” cleavage sites from the manuscript to avoid confusion.

Performing and showing the outcomes of the requested controls will help to understand whether sP and iP really have such a large number of unique cleavage sites (348 of 762 cleavage sites identified). Using longer kinetics will most likely dramatically reduce this number. Even in case that there are only very few differences this would not affect the relevance of the study but will modify one of the main take home messages of the study.

We have performed the experiments requested by the reviewer and feel that they have strengthened the message of the paper. Our results suggest that specificity differences are clearly evident between the two proteasomes. We agree with the reviewer that the altered substrate specificity largely results in the proteasomes displaying differing efficiencies of cleavage for certain substrates. Therefore, we have removed the term “unique” cleavage sites from the paper to avoid confusion as stated above. See also our comment to Figure 4 regarding epitope generation.

I would also suggest to add a supplementary figure comparing the outcome of Figure 1, with those of Toes et al., JEM 2001 and Mishto et al., EurJI 2014 to better understand differences and similarities.

We have now provided a comparison of the results from Figure 1 with those from Toes et al. and Mishto et al. (Figure 1—figure supplement 3).

Figure 2.Figure 2 may serve the purpose of the idea but are not very informative. It is not clear how the experiment was performed. Were these peptides individually processed by either sP or iP and if so, what were the conditions. Are these peptides equally well degraded by both isoforms; are these all the cleavage sites detected by MS?

The legend in Figure 2 has been modified to clarify that the cleavages shown are from the MSP-MS assays associated with Figure 1. The MSP-MS assay conditions are provided in the Materials and methods section. In Figure 2, the time point of first appearance now has been noted for the cleavages used in fluorogenic substrate design to provide a measure of cleavage selectivity. Only cleavages are shown that are common to both biological replicates of each proteasome.

Figure 2—figure supplement 1.There seems to be an editorial error for the km value of EWFW-ACC for i20S. It probably should read 1.8 μm instead 7.0 uM.

The *K*_m_ values provided for the table in Supplementary file 2 are correct. The selectivity of the EWFW-ACC substrate for the iP is driven by the greater *k*_cat_.

Figure 3.It is somewhat surprising that independent of the sP/iP ratio the activity measured with the commonly used substrate Suc-LLVY-AMC remained unchanged. This certainly stands in contrast to several reports by Gaczynska and coworkers (cited) as well as by many others.

The assay was performed using a comparable substrate concentration of 30 μM LLVY where similar iP and cP activity is observed (10 μM EWFW and 30 μM iso-VQA were used by comparison). This has been clarified in the Results section (subsection 2 Fluorogenic Substrates Enable Cellular Activity Profiling of LMP7 and β5”), and substrate concentrations have been provided in the legend for Figure 3 supplemental Michaelis-Menten plot for LLVY (Figure 3—figure supplement 1) also has been added to demonstrate the maximal 2.0-fold iP selectivity observed at *V*_max_.

Figure 4.I do not really understand why the authors state that iP clearly expands the MHC-I repertoire when the authors don´t find proteasome isoform specific cleavage sites which generate the tested epitopes. As documented in Figure 4 the most sP preferred epitope HTQVIEL>ERKFSHQ is also generated by iP, albeit less efficiently. Similarly, the most iP favoured epitope is also generated by sP. My interpretation of the data shown is that there are epitopes which are more efficiently generated by either sP or iP which consequently will affect their presentation at the cell surface. (in this context see Zanker, D., Waithman, J., Yewdell, J. W. and Chen, W.. J. Immunol. 2013. 191: 52-59.)

We agree that the current language about the iP “expanding” the peptide repertoire may be confusing. We have changed this language in the Abstract and throughout the manuscript to state that iP specificity may alter the MHC I peptides displayed due to the different cleavage efficiency of the iP for certain epitopes.

In part the experiments shown also answer my concerns expressed for the data in Figure 1. Here, in Figure 4 the authors have performed longer kinetics as requested above and conclude that many peptides have an equivalent potential of being generated by either proteasome isoform. Even the tryptophan-cleavage site, although preferred by iP is shown to be used by sP as well, albeit less efficiently (Figure 4).This reviewer has some concerns regarding the label-free quantification of the peptides. It is well known that the MS ion signal strongly varies between peptides of different sequence. This is irrelevant with a dataset of thousands of peptides but render not applicable the label-free strategy to quantify a small number of short peptides (Bassani-Sternberg et al., MCP 2015; Liepe et al., Science 2016; Tenzer et al., Nat Immunol 2009).

The reviewer is correct that MS ion signal is strongly dependent upon peptide sequence composition. To clarify, the figure only compares *relative* cleavage rates by the iP and cP for each individual peptide. Therefore, the figure provides a relative ranking of iP versus cP peptide cleavage selectivity against the set of epitopes queried.

In Materials and methods it is stated that 89 epitope derived peptides were analysed but only slightly more the 50 are documented in Figure 5 (please explain).

Only the 56 peptides are shown that were cleaved between positions 7 and 8 (the MHC site) and for which accurate label-free quantification was possible. This information is provided in the Figure 4 legend for clarity.

Figure 5.The authors show that in cells mostly expressing iP, selective inhibition of LMP7 results in an accumulation of ub-conjugates. They infer from these results that iP activity makes a significant contribution to UPS capacity. This conclusion is in agreement with previously published data Seifert et al. Cell (2010), which I would suggest to cite in this context.

We thank the reviewer for the suggestion and have now included the Seifert et al. reference.

That inhibition of the beta5 site doesn´t evoke similar effects is most likely due to the small amounts of sP present in these samples.The authors observe a bounce back of LMP7 activity after irreversible inhibition of LMP7 as well as upregulation of both PSMB8 and PSMB5 mRNAs during the recovery phase. This is a well-known phenomenon. That PSMB5 mRNAs has little impact on β 5 activity is also to be expected because in the presence of both β 5 and LMP7 units, incorporation of LMP7 is dominant due its higher binding affinity for the proteasome maturation protein POMP.

We agree with the reviewer that recovery of LMP7 is indeed a known phenomenon (for example, Lee Br. J. Haematol.; PMID: 27071340). However, to our knowledge, the present study is the first to report the dynamics of LMP7 and beta5 recovery following selective inhibition of the iP (as opposed to CFZ treatment). Dissecting the activities LMP7 and beta5 individually has been made possible by the fluorogenic substrates developed.

Reviewer #2:[…] Different proteasome preparations are being used in the various assays: sometimes SDS-activated and sometimes PA28-activated. To what extent are these methods providing similarly activated proteasomes?

We compared cleavage of LLVY and our lead fluorogenic substrates using both activation methods and found cleavage to be quite similar (Figure 2—figure supplement 4). SDS detergent was not compatible with our MS assay, and therefore, MS-based profiling was performed only with PA28 activation.

Peptides, rather than (unfolded) proteins are being used to establish cleavage preferences. To what extent are peptides suitable protein substitutes? what about concentration, reaction times, other factors that may be different in cells?

We agree that assaying peptides may yield differences compared to intact protein substrates. However, the peptide library used has the advantage of consisting of chemically defined sequences with uniform amino acid distribution. Linking biochemical conditions to cellular contexts is a challenge of any biochemical assay, and we have added the caveats mentioned to the Discussion section for clarity (first paragraph).

Why only beta5c/beta5i?

We have elected to focus on substrate development for the beta5 and LMP7 subunits because these subunits represent the dominant activities of each proteasome, and they have been the major focus of pharmaceutical drug development efforts to date. We agree that substrates could in principle be designed for the remaining catalytic subunits using a similar strategy.

Are ONX0914 and PR825 – developed for murine proteasomes, equally selective against their respective human proteasome targets (given that recent literature point towards differences in the active sites of especially the chymotryptic sites from these two species)?

The compounds ONX-0914 and PR-825 were initially developed against the human proteasome and are cross-reactive with the mouse proteasome. ONX-0914 and PR-825 have similar potency for their target proteasomes (Figure 1—figure supplement 2 in Muchamuel Nat. Med. 2009; PMID:19525961).

Why, in designing selective fluorogenic substrates, not considering non-proteinogenic amino acids, also in view of the literature on likewise-designed inhibitors?

We completely agree that non-proteinogenic amino acids could be an excellent strategy to improve the selectivity of the substrates. With the exception of norleucine, both substrate libraries used in our study consist of proteinogenic amino acids, and therefore our fluorescent substrate design only focused on these amino acids. Several groups, such as the Marcin Drag lab at Wroclaw University of Technology, have developed substrate libraries that use non-proteinogenic amino acids and may use the specificity information revealed here as a starting point for development of improved substrates. However, we feel that the proteinogenic tools developed here have sufficient selectivity to function in cellular lysates and further modification with non-proteinogenic amino acids is beyond the scope of the study.

In using the substrates in mixtures containing both proteasome species: how to correct for minor processing of the 'unwanted' proteasome (since the reagents are not truly selective?)

The reagents developed can be combined with subunit-selective inhibitors to correct for processing by the unwanted proteasome. Continued optimization of subunit-selective inhibitors in parallel will only improve the feasibility of this approach.

Next top cP and iP species mixed proteasomes exist and that may have other substrate preferences: how does the work take the possible existence of such species into account?

We certainly appreciate both the interest in and the challenge of assaying / discriminating mixed proteasomes. It is not well known to what extent the specificity of these species differ. However, we feel that this topic is well beyond the scope of the study because, to our knowledge, the mixed proteasome selectivity of any fluorogenic substrate or small molecule developed to date has not been fully elucidated.

Presentation of the work, I would favour some chemical structures, possibly also a chemical scheme (even though the methodology to synthesise the compounds has been published from the same group some time ago). In general I find the manuscript hard to get into, with lots of abbreviations.

We have now provided chemical structures of the lead substrates (Figure 2) and a synthetic scheme (Figure 2—figure supplement 3) to make the manuscript more approachable for a wider audience.

A survey of the pros and cons of fluorogenic substrates in comparison with other methodology (activity-based protein profiling, the Procise assay) would be helpful.

We have added a survey of the pros and cons of fluorogenic substrates to the Discussion section (first paragraph).

[Editors' note: further revisions were requested prior to acceptance, as described below.]

Reviewer #1:I appreciate the additional work by Winter et al. Their efforts certainly improved the manuscript. In particular, the kinetics experiments and the biological replicates of the MSP-MS assays give now a clearer picture of the results and the potency of the method.Having said this, there still remain some points which in my mind cast doubt onto the robustness of some the experiments.

The main criticism from the reviewer is that we are not sampling a large enough number of cleavages from the library to comment on whether each proteasome truly has unique substrates. To increase the number of cleavages, the reviewer proposes (a) increasing the number of replicates from two to five, (b) comparing ALL and not just cleavages common between replicate assays, and (c) letting the assay run longer until it "reaches a plateau." The number the reviewer suggests is 40-50% of cleavage-site usage (up to 1482 out of 2964 possible cleavage-sites in the 228-member library).

Our Response:

1) The data presented in the paper using the MSP-MS and PS-SCL technologies represent the most extensive extended substrate specificity analysis performed on the iP and the cP to date. Because of the global approach (e.g., breadth) that attempts to comprehensively evaluate the specificity of the enzymes, detailed analysis (e.g., depth) on each of the substrates is sacrificed. This global approach allowed us to develop iP- and cP-selective substrates, which was ultimately one of the goals of our study.

2) The MSP-MS assay detects the most dominant cleavage products in the diverse set of tetradecapeptides when run under Michaelis Menten conditions as we have done in the manuscript. This is best seen in the progress curves provided in Figure 1—figure supplement 1, which demonstrate that the number of new peptide bonds being cleaved in the library by the iP and cP begins to slow after 60 minutes. At 60 minutes, the iP and cP cleaved 264 and 225 peptide bonds corresponding to 8.9% and 7.5% of the 2,964 peptide bonds in the library. After 4 hours of incubation, the total number of new cleavage sites had not yet doubled (498 for the iP and 425 for the cP).

1)

3) As the peptides get smaller due to hydrolysis by the proteasome, they eventually become too small to detect reliably, as now schematically illustrated in Figure 1—figure supplement 2. This is why the assay predominately detects initial cleavage products and running the assay longer will not uncover additional activities.

4) The MSP-MS peptide library has now been used to uncover the substrate specificity of ~80 proteases including enzymes with substrate preferences that are similar to the β1 (caspase-like) and β2 (trypsin-like) subunits of the proteasomes. For example, caspase activity was identified in a recent paper (Julien et al. PNAS 2016; PMID: 27006500). Similarly, tryptic and chymotryptic activities were identified in a separate paper (Winter et al. MBio 2016; PMID: 27624133). Therefore, our assay can detect these same specificities associated with the proteasome. However, since most of the specificity discovered in our study is chymotrypsin-like, we don’t believe that the assay lacks “sensitivity.”

5) The commercial preparations of the iP we used contain a significant portion of the β1 subunit from the cP. However, we are primarily detecting activity associated with the chymotrypsin-like subunit of the iP (LMP7) since it is the major activity of the proteasome. This is consistent with the development of the proteasome inhibitors carfilzomib and bortezomib, which target this subunit. Our assays detect the most dominantly cleaved substrates. However, we are also able to detect other activities as mentioned above.

6) Points: The reviewer misidentifies the false discovery rate (FDR) from the MS experiments. The "0.05" referred to is the maximum expectation value for peptide hits. The peptide FDR from our original paper introducing the MSP-MS assay is <0.17% (O’Donoghue et al. Nat Methods 2012; PMID 23023596). The reviewer also disagrees with the use of the universal LLVY fluorogenic substrate to normalize the proteasome input for the MSP-MS assays. However, our MSP-MS data proves that at the concentration used, LLVY cleavage is an appropriate normalization strategy because we achieve very similar numbers of cleavages for the two proteasomes against our unbiased MSP-MS library using this approach.

7) The sum total of the data presented in the paper suggests that if we were able to run the MSP-MS assay to its theoretical limit, each site cleaved by one proteasome would ultimately be cleaved by the other. For example, from the more sophisticated label-free quantitation in the MSP-MS assay performed with the MHC peptides (Figure 4), it is clear that even highly favored sequences are cleaved by the opposing proteasome at some rate. This is even the case for the optimized fluorogenic substrates. Therefore, our conclusion in the paper is that proteasome substrate specificity preferences (which clearly exist) primarily lead to differences in "quantity" of substrate cleavage and epitope generation, as opposed to strong “qualitative” differences.

Revisions to the Manuscript:

1) The Results section has been modified (subsection “Global Substrate Specificity Screen of the iP and cP”, first paragraph) to emphasize that the MSP-MS assay is being performed as a screening strategy for iP and cP cleavage and that discriminatory peptides have been identified with a particular emphasis on the β5/LMP7 subunits of the proteasome. We have also clarified that the β1 (caspase-like) activity is only a minor contributor to the overall cleavage profile (subsection “Global Substrate Specificity Screen of the iP and cP”, second paragraph and subsection “ProCISE Quantification of Proteasome Subunit Levels and IC50 Values”, last paragraph).

2) The limitations of the MSP-MS assay now have been stated in the Results section (subsection “Global Substrate Specificity Screen of the iP and cP”, first paragraph) to clarify the percentage of peptides that are being cleaved and why running the assay under longer or under other conditions would not provide additional information. We have also added a figure to illustrate this concept (Figure 1—figure supplement 2).

3) We have re-emphasized throughout the manuscript (Abstract, Results, and Discussion) that the substrate specificity differences displayed by the proteasome lead to predominately quantitative differences in substrate cleavage and epitope generation (as opposed to absolute differences in selectivity).

4) As requested by the reviewer, we have deposited the raw mass spectrometry data from the paper (subsection “Multiplex Substrate Profiling by Mass Spectrometry (MSP-MS)”, last paragraph and subsection “MHC I Peptide Library Cleavage Assay”, last paragraph). We have also provided the mass spectrometry peptide reports and sequences for both the MSP-MS library (Figure 1—source data 1) and MHC library (Figure 4—source data 1).